# Differentially Private Generalized Linear Models Revisited

**Raman Arora**
Department of Computer Science
The Johns Hopkins University
arora@cs.jhu.edu

**Raef Bassily**
Department of Computer Science & Engineering
Translational Data Analytics Institute (TDAI)
The Ohio State University
bassily.1@osu.edu

**Cristóbal Guzmán**
Inst. for Mathematical and Comput. Eng.
Fac. de Matemáticas and Esc. de Ingeniería
Pontificia Universidad Católica de Chile
c.guzman@utwente.nl

**Michael Menart**
Department of Computer Science & Engineering
The Ohio State University
menart.2@osu.edu

**Enayat Ullah**
Department of Computer Science
The Johns Hopkins University
enayat@jhu.edu

## Abstract

We study the problem of $(\epsilon, \delta)$-differentially private learning of linear predictors with convex losses. We provide results for two subclasses of loss functions. The first case is when the loss is smooth and non-negative but not necessarily Lipschitz (such as the squared loss). For this case, we establish an upper bound on the excess population risk of $\widetilde{O}\left(\frac{\|w^*\|}{\sqrt{n}} + \min\left\{\frac{\|w^*\|^2}{(n\epsilon)^{2/3}}, \frac{\sqrt{d}\|w^*\|^2}{n\epsilon}\right\}\right)$, where $n$ is the number of samples, $d$ is the dimension of the problem, and $w^*$ is the minimizer of the population risk. Apart from the dependence on $\|w^*\|$, our bound is essentially tight in all parameters. In particular, we show a lower bound of $\widetilde{\Omega}\left(\frac{1}{\sqrt{n}} + \min\left\{\frac{\|w^*\|^{4/3}}{(n\epsilon)^{2/3}}, \frac{\sqrt{d}\|w^*\|}{n\epsilon}\right\}\right)$. We also revisit the previously studied case of Lipschitz losses [SSTT20]. For this case, we close the gap in the existing work and show that the optimal rate is (up to log factors) $\Theta\left(\frac{\|w^*\|}{\sqrt{n}} + \min\left\{\frac{\|w^*\|}{\sqrt{n\epsilon}}, \frac{\sqrt{\mathrm{rank}}\|w^*\|}{n\epsilon}\right\}\right)$, where rank is the rank of the design matrix. This improves over existing work in the high privacy regime. Finally, our algorithms involve a private model selection approach that we develop to enable attaining the stated rates without a-priori knowledge of $\|w^*\|$.

## 1 Introduction

Ensuring privacy of users' data in machine learning algorithms is an important desideratum for multiple domains, such as social and medical sciences. Differential privacy (DP) has become the gold standard of privacy-preserving data analysis as it offers formal and quantitative privacy guarantees and enjoys many attractive properties from an algorithmic design perspective [DR+14]. However, for various basic machine learning tasks, the known risk guarantees are potentially sub-optimal and pessimistic.

36th Conference on Neural Information Processing Systems (NeurIPS 2022).

|  |  | H-Smooth, Non-negative | | | | G-Lipschitz | |
|---|---|---|---|---|---|---|---|
|  |  | $\sqrt{H}\,\|w^*\|\,\|\mathcal{X}\| \leq \|\mathcal{Y}\|$ | | $\sqrt{H}\,\|w^*\|\,\|\mathcal{X}\| > \|\mathcal{Y}\|$ | |  |  |
|  |  | $d \leq \left(\frac{\|w^*\|\sqrt{H}\|\mathcal{X}\|n\epsilon}{\|\mathcal{Y}\|}\right)^{2/3}$ | $d > \left(\frac{\|w^*\|\sqrt{H}\|\mathcal{X}\|n\epsilon}{\|\mathcal{Y}\|}\right)^{2/3}$ | $d \leq (n\epsilon)^{2/3}$ | $d > (n\epsilon)^{2/3}$ | rank $\leq n\epsilon$ | rank $> n\epsilon$ |
| DP | UB | $\frac{\sqrt{H}\|w^*\|\|\mathcal{X}\|\|\mathcal{Y}\|\sqrt{d}}{n\epsilon}$ 
 Theorem 1 | $\frac{\left(\sqrt{H}\|w^*\|\|\mathcal{X}\|\right)^{4/3}\|\mathcal{Y}\|^{2/3}}{(n\epsilon)^{2/3}}$ 
 Theorem 2 | $\frac{H\|w^*\|^2\|\mathcal{X}\|^2\sqrt{d}}{n\epsilon}$ 
 Theorem 1 | $\frac{H\|w^*\|^2\|\mathcal{X}\|^2}{(n\epsilon)^{2/3}}$ 
 Theorem 2, 3 | $\frac{G\|w^*\|\|\mathcal{X}\|\sqrt{\text{rank}}}{n\epsilon}$ 
 [SSTT20] | $\frac{G\|w^*\|\|\mathcal{X}\|}{\sqrt{n\epsilon}}$ 
 Theorem 10, 5 |
| DP | LB | Tight 
 Theorem 4 | Tight 
 Theorem 4 | $\frac{\sqrt{H}\|w^*\|\|\mathcal{X}\|\|\mathcal{Y}\|\sqrt{d}}{n\epsilon}$ 
 Theorem 4 | $\frac{\left(\sqrt{H}\|w^*\|\|\mathcal{X}\|\right)^{4/3}\|\mathcal{Y}\|^{2/3}}{(n\epsilon)^{2/3}}$ 
 Theorem 4 | Tight 
 Theorem 6 | Tight 
 Theorem 6 |
| Non-private | UB | $\frac{\sqrt{H}\|\mathcal{X}\|\|\mathcal{Y}\|\|w^*\|}{\sqrt{n}}$ 
 [SST10] | | | | $\frac{G\|w^*\|\|\mathcal{X}\|}{\sqrt{n}}$ 
 [NY83] | |
| Non-private | LB | $\frac{\min\{\|\mathcal{Y}\|,\|w^*\|\|\mathcal{X}\|\}}{\sqrt{n}} + \min\left\{\|\mathcal{Y}\|^2, \frac{H\|w^*\|^2\|\mathcal{X}\|^2+d\|\mathcal{Y}\|^2}{n}, \frac{\sqrt{H}\|w^*\|\|\mathcal{Y}\|\|\mathcal{X}\|}{\sqrt{n}}\right\}$ 
 [SST10, Sha15] | | | | Tight 
 [NY83] | |

Table 1: Summary of Rates. Parameters: $d$: dimension, $n$: sample size, $H$: smoothness parameter, $w^*$: minimum norm population risk minimizer, $\|\mathcal{X}\|$: bound on feature vectors, $\|\mathcal{Y}\|$: bound on loss at zero, $G$: Lipschitzness parameter, rank: expected rank of the design matrix, $\epsilon$: privacy parameter ($\delta$ factors omitted). The actual private excess risk bounds are the sum of the expressions shown in the DP rows and their non-private counterparts. Details on non-private lower bounds in Appendix E.

In this work, we make progress towards resolving one of the most basic machine learning problems under differential privacy: learning linear predictors with convex losses.

## 1.1 Related Work

Differentially private machine learning has been thoroughly studied for over a decade. In the Lipschitz-convex setting, tight rates are known for both the empirical and population risk [BST14, BFTGT19]. Specifically, it was shown that in the constrained setting, dependence on the dimension in the form of $\Omega\left(\frac{\sqrt{d}}{n\epsilon}\right)$ is unavoidable even for generalized linear models (GLM) (see Section 2 for a formal definition). By contrast, in the unconstrained setting, it has been shown that dimension independent rates are possible for GLMs [JT14]. In this setting, assuming prior knowledge of $\|w^*\|$, the best known rate is $O\left(\frac{\|w^*\|}{\sqrt{n}} + \frac{\|w^*\|\sqrt{\text{rank}}}{n\epsilon}\right)$ [SSTT20], where rank is the expected rank of the design matrix. However, without prior knowledge of $\|w^*\|$, these methods exhibit quadratic dependence on $\|w^*\|$. Furthermore, these results crucially rely on the assumption that the loss is Lipschitz to bound the sensitivity. Although gradient clipping has been proposed to remedy this problem [SSTT20, CWH20], it is known that the solution obtained by clipping may not coincide with the one of the original model.

Without Lipschitzness, work on differentially private GLMs has largely been limited to linear regression [Wan18, CWZ21]. Here, dimension independent rates have only been obtained under certain sparsity assumptions.

More generally, smooth non-negative losses have been studied in the non-private setting by [SST10], where it was shown such functions can obtain risk guarantees with linear dependence on the minimizer norm (as in the Lipschitz case). This work also established a lower bound of $\Omega\left(\frac{1}{\sqrt{n}}\right)$ on the excess population risk for this class of loss functions. [Sha15] additionally establishes a lower bound of $\Omega\left\{\min\left\{\frac{\|w^*\|^2+d}{n}, \frac{\|w^*\|}{\sqrt{n}}\right\}\right\}$ on the excess population risk by way of linear regression[1].

## 1.2 Our Contributions

**Smooth nonegative GLMs.** Our primary contribution is a new and nearly optimal rate for the problem of differentially private learning of smooth GLMs. In this setting, we focus on characterizing the excess risk in terms of $n, d, \epsilon$ and $\|w^*\|$. Specifically, we show that it is possible to achieve a rate of $\widetilde{O}\left(\frac{\|w^*\|}{\sqrt{n}} + \min\left\{\frac{\|w^*\|^2}{(n\epsilon)^{2/3}}, \frac{\sqrt{d}\|w^*\|^2}{n\epsilon}\right\}\right)$ on the excess population risk. Our new rates exhibit an interesting low/high dimensional transition at $d \approx (\|w^*\|\,n\epsilon)^{2/3}$. First, in the low dimensional regime, we develop a novel analysis of noisy gradient decent (GD) inspired by techniques from [SST10]. In particular, we show that Noisy GD gives an improved rate for non-negative smooth functions (not necessarily GLMs). This is based on an average stability analysis of Noisy GD. As we elaborate in

---

[1]The [SST10] bound assumes $\|\mathcal{Y}\|, H, \|\mathcal{X}\| = \Omega(1)$. The bounds of [Sha15] were originally stated for the constrained setting, but can easily be converted. More details in Appendix E.

Section 3.1, a straightforward application of uniform stability leads to sub-optimal bounds and hence a new analysis is required. We note in passing that this upper bound works for (unconstrained) DP-SCO with smooth (non-Lipschitz) losses, which is of independent interest. For the high dimensional regime, we perform random projections of the data (specifically, the Johnson-Lindenstrauss transform) for dimensionality reduction, roughly reducing the problem to its low dimensional counterpart. We also develop a lower bound for the excess risk under DP of $\widetilde{\Omega}\left(\min\left\{\frac{\|w^*\|^{4/3}}{(n\epsilon)^{2/3}}, \frac{\sqrt{d}\|w^*\|}{n\epsilon}\right\}\right)$. We note that non-privately a lower bound of $\widetilde{\Omega}\left(\frac{1}{\sqrt{n}} + \min\left\{\frac{d+\|w^*\|^2}{n}, \frac{\|w^*\|}{\sqrt{n}}\right\}\right)$ is known on the excess population risk [SST10, Sha15]. We note that these private and non-private lower bounds imply that our bound is optimal up to factors of $\|w^*\|$ (see Table 1).

**Lipschitz GLMs.** For the Lipschitz case, we close a subtle but important gap in existing rates. In this setting, it has been shown that one can characterize the excess risk in terms of the expected rank of the design matrix, rank, instead of $d$ [SSTT20]. In this setting, the best known rate was $\widetilde{O}\left(\frac{\|w^*\|}{\sqrt{n}} + \frac{\sqrt{\text{rank}}\|w^*\|}{n\epsilon}\right)$. We show an improved rate of $\widetilde{O}\left(\frac{\|w^*\|}{\sqrt{n}} + \min\left\{\frac{\|w^*\|}{\sqrt{n\epsilon}}, \frac{\sqrt{\text{rank}}\|w^*\|}{n\epsilon}\right\}\right)$. This improves in the high privacy regime where $\epsilon \leq \frac{\text{rank}}{n}$. In fact, the upper bound $O\left(\frac{\|w^*\|}{\sqrt{n\epsilon}}\right)$ for this rate can be obtained with only minor adjustments to the regularization method of [JT14]. Our second contribution in this setting is extending the lower bound of [SSTT20] to hold for all values of $\|w^*\| > 0$ and rank $\in [n]$. This is in contrast to the original lower bound which only holds for problem instances where $\|w^*\|^2 = \text{rank}$ and rank $\in [n\epsilon]$.

**Model selection.** As part of our methods, we develop a differentially private model selection approach which eliminates the need for a-priori knowledge of $\|w^*\|$. Although such methods are well established in the non-private case, (see e.g. [SSBD14]), in the private case no such methods have been established. Our method, as in the non-private case, performs a grid search over estimates of $\|w^*\|$ and picks the best model based on the loss. However, in the private setting we must account for the fact the the loss evaluation must be privatized. This is non-trivial in the non-Lipschitz smooth case as the loss at a point $w$ may grow quadratically with $\|w\|$.

**Lower bounds for Non-Euclidean DP-SCO.** Our lower bound construction generalizes to Non-Euclidean $\ell_p/\ell_q$ variants of DP-SCO with Lipschitz convex losses [BGN21]. Herein, we assume that the loss function is $G_q$-Lipschitz with respect to $\ell_q$ norm, and radius of the constraint set is bounded in $\ell_p$ norm by $B_p$. For this setting, we give a lower bound of $\Omega\left(G_q B_p \min\left(\frac{1}{(n\epsilon)^{1/p}}, \frac{d^{(p-1)/p}}{n\epsilon}\right)\right)$ on excess empirical/population risk of any (potentially unconstrained) $(\epsilon, \delta)$-DP algorithm; see Corollary 4 in Appendix B.4 for a formal statement and proof. For $p = \infty$ and $p \geq 2, d \leq n\epsilon$, this matches best known upper bounds in [BGN21].

**Non-private settings.** As by-products, we give the following new results for the non-private setting. For details on the parameters used below we refer to Table 1.

1. We show that gradient descent, when run on convex non-negative $\widetilde{H}$ smooth functions (not necessarily GLMs), it achieves the optimal rate of $O\left(\frac{\sqrt{\widetilde{H}}\|w^*\|\|\mathcal{Y}\|}{\sqrt{n}}\right)$ (see Corollary 1). This is done via an average-stability analysis of gradient descent. This result is interesting as it also shows GD only needs $n$ iterations, which is known not to work for non-smooth SCO [BFGT20, ACKL21, AKL21].

2. In Section D, we give a procedure to boost the confidence of algorithms for risk minimization with convex non-negative $\widetilde{H}$ smooth functions (not necessarily GLMs). The standard boosting analysis based on Hoeffding's inequality does not give a bound with a linear dependence on the parameters $(\|w^*\|, \|\mathcal{X}\|, \|\mathcal{Y}\|)$, and hence a tighter analysis is required.

### 1.3 Techniques

**Upper bounds.** We give two algorithms for both the smooth and Lispschitz cases. The first method is simple and has two main steps. First, optimize the regularized empirical risk over the constraint set $\{w : \|w\| \leq B\}$ for some $B \geq \|w^*\|$. Then output a perturbation of the regularized minimizer

with Gaussian noise (which is not requried to be in the constraint set). This method is akin to that of [JT14] with the modification that the regularized minimizer is constrained to a ball. We elaborate on this key difference shortly.

The second method is based on dimensionality reduction. We use smaller dimensional data-oblivious embeddings of the feature vectors. A linear JL transform suffices to give embeddings with the required properties. We then run a constrained DP-SCO method (Noisy GD) in the embedded space, and use the transpose of the JL transform to get a $d$ dimensional model. In this method, the embedding dimension required is roughly the threshold on dimension at which the rates switch from dimension dependent to independent bounds. We also remark that [NêUZ20] applied a similar technique to provide dimension independent classification error guarantees for privately learning support vector machines under hard margin conditions.

We note that a crucial part in all of these methods is the use of constrained optimization as a subroutine, where the constraint set is a ball of radius $\|w^*\|$. This is in stark contrast to the Lipschitz case where existing methods such as those presented by [JT14, SSTT20] rely on the fact that projection is not required. In the smooth case however, constrained optimization helps ensure that the norm of the gradient is roughly bounded by the diameter of the constraint set. We note that in the high dimensional regime, the property that the *final* output of the algorithm can have large norm is still crucial to the success of our algorithms.

**Lower bounds.** For our lower bounds in the smooth case we rely on the connection between stability and privacy. Specifically, we will utilize a lemma from [CH12] which bounds the accuracy of one-dimensional differentially private algorithms. We then combine this with packing arguments to obtain stronger lower bounds for high dimensional problems. For the Lipschitz case, we adapt the method of [SSTT20].

## 2 Preliminaries

In the following we detail several concepts needed for the presentation of this paper.

**Risk minimization.** Let $\mathcal{X} \subseteq \mathbb{R}^d$ be the domain of features, and $\mathcal{Y} \subseteq \mathbb{R}$ be the domain of responses. A linear predictor is any $w \in \mathbb{R}^d$. Let $\ell : \mathbb{R}^d \times (\mathcal{X} \times \mathcal{Y}) \to \mathbb{R}$ be a loss function. Given some unknown distribution $\mathcal{D}$ over $(\mathcal{X} \times \mathcal{Y})$, we define the population loss $L(w; \mathcal{D}) = \underset{(x,y) \sim \mathcal{D}}{\mathbb{E}} [\ell(w; (x, y))]$.

Given some dataset $S \in (\mathcal{X} \times \mathcal{Y})^n$ drawn i.i.d. from $\mathcal{D}$, the objective is to obtain $\widehat{w} \in \mathbb{R}^d$ which minimizes the excess population risk, $L(\widehat{w}; \mathcal{D}) - \underset{w \in \mathbb{R}^d}{\min} \{L(w; \mathcal{D})\}$. Given a population risk minimization problem, we will denote $w^*$ to be a *minimum norm* solution to this problem. We define the empirical risk as $\widehat{L}(w; S) = \frac{1}{n} \underset{(x,y) \in S}{\sum} \ell(w; (x, y))$. We define the following quantities for notational convenience: $\varepsilon_{\mathsf{risk}}(w) = L(w; \mathcal{D}) - L(w^*; \mathcal{D})$, $\varepsilon_{\mathsf{erm}}(w) = \widehat{L}(w; S) - \widehat{L}(w^*; S)$, and $\varepsilon_{\mathsf{gen}}(w) = L(w; \mathcal{D}) - \widehat{L}(w; S)$. We define $\mathcal{B}_B$ to be the Euclidean ball of radius $B$ on $\mathbb{R}^d$.

**Generalized linear models.** We will more specifically be interested in the problem of learning generalized linear models, where there exists some function $\phi : \mathbb{R} \times \mathcal{Y} \to \mathbb{R}$ such that the loss function can be written as $\ell(w; (x, y)) = \phi(\langle w, x \rangle, y)$. We define parameter bounds $\|\mathcal{X}\| = \max_{x \in \mathcal{X}} \|x\|$ and $\|\mathcal{Y}\|^2 = \max_{y \in \mathcal{Y}} |\phi(0, y)|$. Note that for many common loss functions, the latter condition is the moral equivalent of assuming labels bounded by $\|\mathcal{Y}\|$. For ease of notation, we write $\phi(\langle w, x \rangle, y)$ as $\phi_y(\langle w, x \rangle)$ and denote function $\phi_y : z \mapsto \phi(z, y)$. We say that the loss function is $G$-Lipschitz GLM if all $y \in \mathcal{Y}$, $\phi_y : \mathbb{R} \to \mathbb{R}$ is $G$-Lipschitz. We similarly define $H$-smooth GLM.

**Differential privacy [DKM+06].** We restrict our investigation to the class of algorithms which minimize the excess population risk under the constraint of differential privacy. A randomized algorithm $\mathcal{A}$ is said to be $(\epsilon, \delta)$ differentially private (i.e., $(\epsilon, \delta)$-DP) if for any pair of datasets $S$ and $S'$ differing in one point and any event $\mathcal{E}$ in the range of $\mathcal{A}$ it holds that

$$\mathbb{P}[\mathcal{A}(S) \in \mathcal{E}] \le e^\epsilon \mathbb{P}[\mathcal{A}(S') \in \mathcal{E}] + \delta.$$

For our lower bounds, we will make use of the following Lemma from [CH12].

**Lemma 1.** *Let $\mathcal{Z}$ be a data domain and let $S$ and $S'$ be two datasets each in $\mathcal{Z}^n$ that differ in at most $\Delta$ entries, and let $\mathcal{A} : \mathcal{Z}^n \to \mathbb{R}$ be any $(\epsilon, \delta)$-DP algorithm. For all $\tau \in \mathbb{R}$, if $\Delta \leq \frac{\log(1/2\gamma)}{\epsilon}$ and $\delta \leq \frac{1}{16}(1 - e^{-\epsilon})$, then $\mathbb{E}\left[|\mathcal{A}(S) - \tau| + |\mathcal{A}(S') - \tau'|\right] \geq \frac{1}{4}|\tau - \tau'|$.*

Finally, we introduce the Johnson-Lindenstrauss (JL) transform to perform random projections.

**Definition 1** (($\alpha, \beta$)-JL property). *A distribution over matrices $\mathbb{R}^{k \times d}$ satisfies $(\alpha, \beta)$-JL property if for any $u, v \in \mathbb{R}^d$, $\mathbb{P}\left[|\langle \Phi u, \Phi v \rangle - \langle u, v \rangle| > \alpha \|u\| \|v\|\right] \leq \beta$.*

It is well known that several such "data-oblivious" (i.e. independent of $u,v$) distributions exist with $k = O\left(\frac{\log(1/\beta)}{\alpha^2}\right)$ [Nel11]. We note that the JL property is typically described as approximation of norms (or distances), but it is easy to deduce the above dot product preservation property from it; for completeness we give this as Lemma 12. Finally, we use $\Phi\mathcal{D}$ to denote the push-forward measure of the distribution $\mathcal{D}$ under the map $\Phi : (x, y) \mapsto (\Phi x, y)$. Similarly, given a data set $S = \{(x_i, y_i)\}_i$, we define $\Phi S := \{(\Phi x_i, y_i)_i\}$

# 3 Smooth Non-negative GLMs

For smooth non-negative GLMs, we present new upper and lower bounds on the excess risk. For our upper bounds, we here assume that the algorithm is given access to some upper bound on $\|w^*\|$, that we denote by $B$. We later show in Section 5 how to obtain such a rate without prior knowledge of $\|w^*\|$. We also emphasize that the privacy of these algorithms holds regardless of whether or not $B \geq \|w^*\|$.

## 3.1 Upper Bounds

Before presenting our algorithms, we highlight some key ideas underlying all our methods. A crucial property of non-negative smooth loss functions which allows one to bound sensitivity, and thus ensure privacy, is the self-bounding property (e.g. Sec. 12.1.3 in [SSBD14]), which states that for an $\widetilde{H}$-smooth non-negative function $f$ and $u \in \mathrm{dom}(\mathrm{f})$, $\|\nabla f(u)\| \leq \sqrt{4\widetilde{H}f(u)}$.

This property implies that the gradient grows at most linearly with $\|w\|$. More precisely, we have the following.

**Lemma 2.** *Let $\ell$ be an $H$-smooth non-negative GLM. Then for any $w \in \mathcal{B}_B$ and $(x, y) \in (\mathcal{X} \times \mathcal{Y})$ we have $\|\nabla\ell(w, (x, y))\| \leq 2\|\mathcal{Y}\|\sqrt{H}\|\mathcal{X}\| + 2HB\|\mathcal{X}\|^2$.*

In order to leverage this property, all our algorithms in this setting utilize constrained optimization as a subroutine, where the constraint set is $\mathcal{B}_B$ and the Lipschitz constant is $G = 2\|\mathcal{Y}\|\sqrt{H}\|\mathcal{X}\| + 2HB\|\mathcal{X}\|^2$. This, in conjunction with the self-bounding property ensures reasonable bounds on sensitivity. In turn, this allows us to ensure privacy without excessive levels of noise.

Finally, we note that our upper bounds for smooth GLMs distinguish low and high dimensional regimes, transitioning at $d = \min\left(\left(\frac{B\sqrt{H}\|\mathcal{X}\|}{\|\mathcal{Y}\|}\right)^{\frac{2}{3}}, 1\right)(n\epsilon)^{\frac{2}{3}}$.

### 3.1.1 Low dimensional regime

We start with the low dimensional setting where we use techniques developed for constrained DP-SCO for Lipschitz losses (not necessarily GLMs). Existing private algorithms for DP-SCO (e.g., [BFTGT19, FKT20]) lead to excess risk bounds that scale with $\frac{HB^2}{\sqrt{n}}$. On the other hand, the optimal non-private rate [SST10] scales with $\frac{\sqrt{H}B}{\sqrt{n}}$, which may indicate that the private rate implied by the known methods is sub-optimal. We show that this gap can be closed by a novel analysis of private GD.

A standard proof of excess risk of (noisy) gradient descent for smooth convex functions is based on uniform stability [HRS15, BFTGT19]. However, this still leads to sub-optimal rates. Hence we turn to an average stability based analysis of GD, yielding the following result.

**Theorem 1.** *Let $\ell$ be a non-negative, convex, $\widetilde{H}$-smooth loss function, bounded at zero by $\|\mathcal{Y}\|^2$. Let $B, n > 0$, $n_0 = \frac{\widetilde{H}B^2}{\|\mathcal{Y}\|^2}$, $G = 2\|\mathcal{Y}\|\sqrt{\widetilde{H}} + 2\widetilde{H}B$. Then, for any $\epsilon, \delta > 0$, Algorithm 1 invoked with $\mathcal{W} = \mathcal{B}_B$, $T = n$, $\sigma^2 = \frac{8G^2 T \log(1/\delta)}{n^2\epsilon^2}$, $\eta = \min\left(\frac{B}{\sqrt{T}\max\left(\sqrt{\widetilde{H}}\|\mathcal{Y}\|, \sigma\sqrt{d}\right)}, \frac{1}{4\widetilde{H}}\right)$ is $(\epsilon, \delta)$-differentially private. Further, given a dataset $S$ of $n \geq n_0$ i.i.d samples from an unknown distribution $\mathcal{D}$, the excess risk of output of Algorithm 1 is bounded as,*

$$\mathbb{E}[\varepsilon_{\mathsf{risk}}(\widehat{w})] \leq O\left(\frac{\sqrt{\widetilde{H}}B\|\mathcal{Y}\|}{\sqrt{n}} + \frac{GB\sqrt{d\log(1/\delta)}}{n\epsilon}\right).$$

We note that since $H$-smooth GLMs satisfy the Theorem condition with parameter $\widetilde{H} \leq H\|\mathcal{X}\|^2$, we obtain results for GLMs as a direct corollary.

**Non-private risk bound:** As a corollary (see Corollary 1 in Appendix A.3.1), with no privacy constraint, the above result (setting $\epsilon \to \infty$ and $\delta = 1$) shows that gradient descent achieves the optimal excess risk bound, previously shown to be achievable by regularized ERM and one-pass SGD [SST10]. The lower bound, $n_0$, simply means that the trivial solution "zero" has larger excess risk.

**Proof sketch of Theorem 1.** The privacy proof simply follows from [BST14] since the loss function is $G$-Lipschitz in the constraint set. For utility, we first introduce two concepts used in the proof. Let $S$ be a dataset of $n$ i.i.d samples $\{(x_i, y_i)\}_{i=1}^n$, $S^{(i)}$ be the dataset where the $i$-th data point is replaced by an i.i.d. point $(x', y')$. Let $\mathcal{A}$ be an algorithm which takes a dataset as input and outputs $\mathcal{A}(S)$. The **average argument stability** of $\mathcal{A}$, denoted as $\varepsilon_{\mathsf{av-stab}}(\mathcal{A})$ is defined as

$$\varepsilon_{\mathsf{av-stab}}(\mathcal{A})^2 = \mathbb{E}_{S,i,(x',y')}\|\mathcal{A}(S) - \mathcal{A}(S^{(i)})\|^2.$$

The **average regret** of gradient descent (Algorithm 1) with iterates $\{w_t\}_{t=1}^T$ is

$$\varepsilon_{\mathsf{reg}}(\mathcal{A}; w^*) = \frac{1}{T}\sum_{j=1}^T \mathbb{E}[\widehat{L}(w_j; S) - \widehat{L}(w^*; S)].$$

The key arguments are as follows: we first bound the generalization error, or on-average stability, in terms of average argument stability and excess empirical risk (Lemma 4). We then bound average argument stability in terms of average regret (Lemma 5). Finally, in Lemma 6, we provide bounds on excess empirical risk and average regret of noisy gradient descent. Substituting these in the excess risk decomposition gives the claimed bound. The full proof with the above lemmas is deferred to Appendix A.3.1.

---

**Algorithm 1** Noisy GD

---

**Input:** Dataset $S$, loss function $\ell$, constraint set $\mathcal{W}$, steps $T$, learning rate $\eta$, noise scale $\sigma^2$
1: $w_0 = 0$
2: **for** $t = 1$ to $T - 1$ **do**
3:    $\xi \sim \mathcal{N}(0, \sigma^2 \mathbb{I})$
4:    $w_{t+1} = \Pi_{\mathcal{W}}\left(w_t - \eta\left(\widehat{L}(w_t; S) + \xi\right)\right)$
     where $\Pi_{\mathcal{W}}$ is the Euclidean projection on to $\mathcal{W}$
5: **end for**
**Output:** $\widehat{w} = \frac{1}{T}\sum_{t=1}^T w_j$

---

### 3.1.2 High dimensional regime

In the high dimensional setting, we present two techniques.

**JL method.** In the JL method, Algorithm 2, we use a data-oblivious JL map $\Phi$ to embed all feature vectors in dataset $S$ to $k < d$ dimensions. Let dataset $\widetilde{S} = \{(\Phi x_i, y_i)\}_{i=1}^n$. We then run projected Noisy GD method (Algorithm 1) on the loss with dataset $\widetilde{S}$ and the diameter of the constraint set as $2B$. Finally, we map the returned output $\widetilde{w}$ back to $d$ dimensions using $\Phi^\top$ to get $\widehat{w} = \Phi^\top\widetilde{w}$. We note that no projection is performed on $\widehat{w}$ and thus the output may have large norm due to re-scaling induced by $\Phi^\top$.

---
**Algorithm 2** JL Method
---
**Input:** Dataset $S = \{(x_1, y_1), ..., (x_n, y_n)\}$, loss function $\ell, k, B, \eta, T, \sigma^2$
  1: Sample JL matrix $\Phi \in \mathbb{R}^{k \times d}$
  2: $\widetilde{S} = \{(\Phi x_1, y_1), (\Phi x_2, y_2), \ldots, (\Phi x_n, y_n)\}$
  3: $\widetilde{w} = \text{NoisyGD}(\widetilde{S}, \ell, \mathcal{B}_{2B}, T, \eta, \sigma^2)$
**Output:** $\widehat{w} = \Phi^\top \widetilde{w}$
---

---
**Algorithm 3** Regularized Output Perturbation
---
**Input:** Dataset $S = \{(x_1, y_1), ..., (x_n, y_n)\}$, loss function $\ell, \lambda, B, \sigma^2$
  1: $\widetilde{w} = \underset{w \in \mathcal{B}_B}{\arg\min} \left\{ L(w; S) + \frac{\lambda}{2} \|w\|^2 \right\}$
  2: $\xi \sim \mathcal{N}(0, \sigma^2 \mathrm{I}_d)$
**Output:** $\widehat{w} = \widetilde{w} + \xi$
---

**Theorem 2.** *Let* $k = O\left( \frac{B\sqrt{H}\|\mathcal{X}\| \log(2n/\delta)n\epsilon}{\|\mathcal{Y}\|\|\mathcal{X}\| + \sqrt{H}B\|\mathcal{X}\|^2} \right)^{2/3}$, $\mathcal{W} = \mathcal{B}_B$, $T = n$, $\sigma^2 = \frac{8G^2 T \log(1/\delta)}{n^2 \epsilon^2}$, $\eta = \min\left( \frac{B}{\sqrt{T} \max\left(\sqrt{H}\|\mathcal{X}\|\|\mathcal{Y}\|, \sigma\sqrt{d}\right)}, \frac{1}{4H\|\mathcal{X}\|^2} \right)$ *and* $n_0 = \frac{HB^2\|\mathcal{X}\|^2}{\|\mathcal{Y}\|^2}$. *Algorithm 2 satisfies* $(\epsilon, \delta)$-*differential privacy. Given a dataset $S$ of $n \geq n_0$ i.i.d samples, of the output $\widehat{w}$ is bounded by*

$$\mathbb{E}[\varepsilon_{\mathsf{risk}}(\widehat{w})] \leq \widetilde{O}\left( \frac{\sqrt{H}B\|\mathcal{X}\|\|\mathcal{Y}\|}{\sqrt{n}} + \frac{\left(\sqrt{H}B\|\mathcal{X}\|\sqrt{\|\mathcal{Y}\|}\right)^{\frac{4}{3}} + \left(\sqrt{H}B\|\mathcal{X}\|\right)^2}{(n\epsilon)^{\frac{2}{3}}} \right).$$

**Proof sketch of Theorem 2.** From the JL property, with $k = O(\log(n/\delta)/\alpha^2)$, w.h.p. norms of all feature vectors and $w^*$, as well as inner products between are preserved upto an $\alpha$ tolerance (see Definition 1). The preservation of norms of feature vectors implies the gradient norms are preserved, and thus privacy guarantee of sub-routine, Algorithm 1, suffices to establish DP. For the utility proof, from our analysis of Noisy GD (Theorem 1) and using the JL property, the excess risk of $\widehat{w}$ under $\mathcal{D}$ w.r.t. the risk of $\Phi w^*$ under $\Phi\mathcal{D}$ is bounded as,

$$\mathbb{E}[L(\widehat{w}; \mathcal{D}) - L(\Phi w^*; \Phi\mathcal{D})] \leq O\left( \frac{\sqrt{H}\|\mathcal{X}\|B\|\mathcal{Y}\|}{\sqrt{n}} + \frac{\left(\sqrt{H}\|\mathcal{X}\|\|\mathcal{Y}\| + H\|\mathcal{X}\|^2 B^2\right) B\sqrt{k \log(1/\delta)}}{n\epsilon} \right).$$

From smoothness and JL property, the, "JL error" is:

$$\mathbb{E}[L(\Phi w^*; \Phi\mathcal{D})] - L(w^*; \mathcal{D})] \leq \widetilde{O}\left( \frac{HB^2\|\mathcal{X}\|^2}{k} \right).$$

The above is optimized for the value of $k$ prescribed in Theorem 2, substituting which gives the claimed bound.

**Constrained regularized ERM + output perturbation.** Our second technique is *constrained* regularized ERM with output perturbation (Algorithm 3). A similar technique for the Lipschitz case was seen in [JT14], however we note that the addition of the constraint set $\mathcal{B}_B$ is crucial in bounding the sensitivity in the smooth case.

**Theorem 3.** *Let* $n_0 = \frac{HB^2\|\mathcal{X}\|^2}{\|\mathcal{Y}\|^2}$. *Then Algorithm 3 run with* $\sigma^2 = O\left( \frac{\left(\|\mathcal{Y}\|^2 + H^2 B^2 \|\mathcal{X}\|^4\right) \log(1/\delta)}{\lambda^2 n^2 \epsilon^2} \right)$ *and* $\lambda = \left( \frac{\left(\|\mathcal{Y}\| + HB\|\mathcal{X}\|^2\right)\sqrt{H}\|\mathcal{X}\|}{Bn\epsilon} \right)^{2/3} (\log(1/\delta))^{1/3}$ *satisfies* $(\epsilon, \delta)$-*differential privacy. Given a dataset $S$ of $n \geq n_0$ i.i.d samples, the excess risk of its output $\widehat{w}$ is bounded as*

$$\mathbb{E}[\varepsilon_{\mathsf{risk}}(\widehat{w})] \leq \widetilde{O}\left( \frac{\sqrt{H}B\|\mathcal{X}\|\|\mathcal{Y}\| + \|\mathcal{Y}\|^2}{\sqrt{n}} + \frac{\left(\sqrt{H}B\|\mathcal{X}\|\right)^{4/3}\|\mathcal{Y}\|^{2/3} + \left(\sqrt{H}B\|\mathcal{X}\|\right)^2}{(n\epsilon)^{2/3}} \right).$$

We note that we can use the same technique in the low dimensional setting too, yielding a rate of $\frac{\sqrt{H}B\|\mathcal{X}\|\|\mathcal{Y}\|+\|\mathcal{Y}\|^2}{\sqrt{n}} + \frac{GB\sqrt{d}}{n\epsilon}$. However, in contrast to Theorem 1 and 2, these results have an additional $\frac{\|\mathcal{Y}\|^2}{\sqrt{n}}$ term (in both regimes). Thus, in the regime when $\|\mathcal{Y}\| \leq \sqrt{H}B\|\mathcal{X}\|$, the two upper bounds are of the same order.

**Proof sketch of Theorem 3.** The privacy proof follows the Gaussian mechanism guarantee together with the fact that the $\ell_2$-sensitivity of constrained regularized ERM is $O\left(\frac{G}{\lambda n}\right)$ [BE02]. For utility, we use the Rademacher complexity based result of [SST10] to bound the generalization error of $\widetilde{w}$. The other term, error from noise, $\mathbb{E}[L(\widehat{w}; \mathcal{D}) - L(\widetilde{w}; \mathcal{D})] \leq O(H\sigma^2\|\mathcal{X}\|^2)$ from smoothness of GLM. Combining these two and optimizing for $\lambda$ gives the claimed result.

## 3.2 Lower Bounds

The proof technique for our lower bound relies on the connection between differential privacy and sensitivity shown in [CH12]. The underlying mechanisms behind the proof is also similar in nature to the method seen in [SU17]. We note that although we present the following theorem for empirical risk, a reduction found in [BFTGT19] implies the following result holds for population risk as well (up to log factors).

**Theorem 4.** *Let $\epsilon \in [1/n, 1]$, $\delta \leq \frac{1}{16}(1 - e^{-\epsilon})$. For any $(\epsilon, \delta)$-DP algorithm, $\mathcal{A}$, there exists a dataset $S$ with empirical minimizer of norm $B$ such that the excess empirical risk of $\mathcal{A}$ is lower bounded by $\Omega\left\{\min\left\{\|\mathcal{Y}\|^2, \frac{(B\|\mathcal{X}\|)^{4/3}(H\|\mathcal{Y}\|)^{2/3}}{(n\epsilon)^{2/3}}, \frac{\sqrt{d}B\|\mathcal{X}\|\|\mathcal{Y}\|\sqrt{H}}{n\epsilon}\right\}\right\}$.*

The problem instance used in the proof is the squared loss function, and thus this result holds additionally for the more specific case of linear regression. We also note this lower bound implies our upper bound is optimal whenever $B\|\mathcal{X}\| \leq \|\mathcal{Y}\|$, which is a commonly studied regime in linear regression [SST10, KL15]. We here provide a proof sketch and defer the full proof to Appendix A.7.

**Proof sketch of Theorem 4** Define $F(w; S) = \frac{1}{n}\sum_{(x,y)\in S}(\langle w, x\rangle - y)^2$. Let $d' < \min\{n, d\}$ and $b, p \in [0, 1]$ be parameters to be chosen later. For any $\boldsymbol{\sigma} \in \{\pm 1\}^{d'}$, define the dataset $S_{\boldsymbol{\sigma}}$ which consists of the union of $d'$ subdatasets, $S_1, ..., S_{d'}$ given as follows. Set $\frac{pn}{d'}$ of the feature vectors in $S_j$ as $\|\mathcal{X}\|e_j$ (the rescaled $j$'th standard basis vector) and the rest as the zero vector. Set $\frac{pn}{2d}(1+b)$ of the labels as $\boldsymbol{\sigma}_j\|\mathcal{Y}\|$ and $\frac{pn}{2d}(1-b)$ labels as $-\boldsymbol{\sigma}_j\|\mathcal{Y}\|$. Let $w^{\boldsymbol{\sigma}} = \arg\min_{w\in\mathbb{R}^d}\{F(w; S_{\boldsymbol{\sigma}})\}$ be the ERM minimizer of $F(\cdot; S_{\boldsymbol{\sigma}})$. Following from Lemma 2 of [Sha15] we have that for any $\bar{w} \in \mathbb{R}^d$ that

$$F(\bar{w}; S_{\boldsymbol{\sigma}}) - F(w^{\boldsymbol{\sigma}}; S_{\boldsymbol{\sigma}}) \geq \frac{p\|\mathcal{X}\|^2}{2d'}\sum_{j=1}^{d'}(\bar{w}_j - w_j^{\boldsymbol{\sigma}})^2. \tag{1}$$

We will now show lower bounds on the per-coordinate error. Consider any $\boldsymbol{\sigma}$ and $\boldsymbol{\sigma}'$ which differ only at index $j$ for some $j \in [d']$. Note that the datasets $S_{\boldsymbol{\sigma}}$ and $S_{\boldsymbol{\sigma}'}$ differ in $\Delta = \frac{pn}{2d'}[(1+b)-(1-b)] = \frac{pbn}{d'}$ points. Let $\tau = w_j^{\boldsymbol{\sigma}} = \frac{\|\mathcal{Y}\|b}{\|\mathcal{X}\|}$ and $\tau' = w_j^{\boldsymbol{\sigma}'} = -\frac{\|\mathcal{Y}\|b}{\|\mathcal{X}\|}$ (i.e. the $j$ components of the empirical minimizers for $S$ and $S_j'$ respectively). Note that $|w_j^{\boldsymbol{\sigma}} - w_j^{\boldsymbol{\sigma}'}| = \frac{2\|\mathcal{Y}\|b}{\|\mathcal{X}\|}$. Setting $d' = \left(\frac{p\|\mathcal{X}\|Bn\epsilon}{\|\mathcal{Y}\|}\right)^{2/3}$ and $b = \left(\frac{\|\mathcal{X}\|B}{\|\mathcal{Y}\|\sqrt{pn\epsilon}}\right)^{2/3}$ ensures $\Delta \leq \frac{1}{\epsilon}$, and thus Lemma 1 can be used to obtain

$$\mathbb{E}\left[|\mathcal{A}(S_{\boldsymbol{\sigma}})_j - w_j^{\boldsymbol{\sigma}}|^2 + |\mathcal{A}(S_{\boldsymbol{\sigma}'})_j - w_j^{\boldsymbol{\sigma}'}|^2\right] \geq \frac{1}{32}\frac{\|\mathcal{Y}\|^2 b^2}{\|\mathcal{X}\|^2} = \frac{B^{4/3}\|\mathcal{Y}\|^{2/3}}{32(\|\mathcal{X}\|pn\epsilon)^{2/3}}.$$

One can now show via a packing argument that

$$\sup_{\boldsymbol{\sigma}\in\{\pm 1\}^{d'}}\left\{\mathbb{E}\left[F(\mathcal{A}(S_{\boldsymbol{\sigma}})) - F(w^{\boldsymbol{\sigma}})\right]\right\} \geq \frac{(\|\mathcal{X}\|B)^{4/3}\|\mathcal{Y}\|^{2/3}p^{1/3}}{128(n\epsilon)^{2/3}},$$

The result then follows from setting $p = \min\left\{1, \frac{d^{3/2}\|\mathcal{Y}\|}{B\|\mathcal{X}\|n\epsilon}\right\}$.

# 4 Lipschitz GLMs

In the Lipschitz case, we close a subtle gap in existing rates. We recall that in this setting a more precise characterization in terms of the expected rank of the design matrix is possible (as opposed to using $d$). The best known upper bound is $\widetilde{O}\left(\frac{\|w^*\|\sqrt{\text{rank}}}{n\epsilon}\right)$ assuming knowledge of $\|w^*\|$. This bound was shown to be optimal when $\epsilon \geq \text{rank}/n$ and $\|w^*\| = \sqrt{\text{rank}}$ [SSTT20].

We first show that in the high privacy regime where $\epsilon \leq \text{rank}/n$, an improved rate is possible. Specifically, we show that in this regime constrained regularized ERM with output perturbation and achieves the optimal rate. In fact, we note that the method of [JT14] (i.e. *unconstrained* regularized ERM with output perturbation), can obtain this rate when $\epsilon = O(1)$ if the regularization parameter is set differently. We present the constrained version in order to leverage Rademacher complexity arguments and provide a slightly cleaner bound that holds for all $\epsilon > 0$.

**Theorem 5.** *Algorithm 3 run with parameters* $\sigma^2 = \frac{4G^2\|\mathcal{X}\|^2 \log(1/\delta)}{\lambda^2 n^2 \epsilon^2}$ *and* $\lambda = \frac{G\|\mathcal{X}\|(\log(1/\delta))^{1/4}}{B\sqrt{n\epsilon}}$ *satisfies* $(\epsilon, \delta)$*-DP. Given a dataset of $n$ i.i.d. samples from $\mathcal{D}$, its output has excess risk* $\mathbb{E}\left[\varepsilon_{\text{risk}}(\widehat{w})\right] = \widetilde{O}\left(\frac{GB\|\mathcal{X}\|}{\sqrt{n}} + \frac{GB\|\mathcal{X}\|}{\sqrt{n\epsilon}}\right)$.

We also state and prove a similar bound using the JL technique in Appendix B.

Next, we generalize the lower bound of [SSTT20] to show this new bound is optimal for all settings of $B$, rank, and $\epsilon$. We now show that a modification of the lower bound present in [SSTT20] shows our upper bound is tight. We note that their lower bound only held for problem instances where $\text{rank} = \|w^*\|^2$ and $\epsilon \leq \text{rank}/n$. By contrast, the upper bound $O(\frac{\sqrt{\text{rank}}\|w^*\|}{n\epsilon})$ holds for any values of rank and $\|w^*\|$.

**Theorem 6.** *Let* $G, \|\mathcal{Y}\|, \|\mathcal{X}\|, B > 0$, $\epsilon \leq 1.2$ *and* $\delta \leq \epsilon$. *For any* $(\epsilon, \delta)$*-DP algorithm* $\mathcal{A}$, *there exists a $G$-Lipschitz GLM loss bounded at zero by $\|\mathcal{Y}\|$ and a distribution $\mathcal{D}$ with $\|w^*\| \leq B$ such that the output of $\mathcal{A}$ on $S \sim \mathcal{D}^n$ satisfies* $\mathbb{E}[\varepsilon_{\text{risk}}(\mathcal{A}(S))] = \Omega\left(GB\|\mathcal{X}\|\min\left(1, \frac{1}{\sqrt{n\epsilon}}, \frac{\sqrt{\text{rank}}}{n\epsilon}\right)\right)$.

All proofs for this section are deferred to Appendix B.

# 5 Adapting to $\|w^*\|$

Our method for privately adapting to $\|w^*\|$ is given in Algorithm 4. We start by giving a high level overview and defining some necessary preliminaries. The algorithm works in the following manner. First we define a number of "guesses" $K$ for $\|w^*\|$, $B_1, ..., B_K$ where $B_j = 2^j : \forall j \in [K]$. Then given black box access to a DP optimization algorithm, $\mathcal{A}$, Algorithm 4 generates $K$ candidate vectors $w_1, ..., w_K$ using $\mathcal{A}$, training set $S_1 \in (\mathcal{X} \times \mathcal{Y})^{n/2}$, and the guesses $B_1, ..., B_K$. We assume $\mathcal{A}$ satisfies the following accuracy assumption for some confidence parameter $\beta > 0$.

**Assumption 1.** *There exists a function* $\text{ERR} : \mathbb{R}^+ \mapsto \mathbb{R}^+$ *such that for any* $B \in \mathbb{R}^+$, *whenever* $B \geq \|w^*\|$, *w.p. at least* $1 - \frac{\beta}{4K}$ *under the randomness of* $S_1 \sim \mathcal{D}^{\frac{n}{2}}$ *and* $\mathcal{A}$ *it holds that* $\varepsilon_{\text{risk}}(\mathcal{A}(S_1, B); \mathcal{D}) \leq \text{ERR}(B)$.

After generating the candidate vectors, the goal is to pick guess with the smallest excess population risk in a differentially private manner using a validation set $S_2$. The following assumption on $\mathcal{A}$ allows us both to ensure the privacy of the model selection algorithm and verify that $\widehat{L}(w_j; S_2)$ provides a tight estimate of $L(w_j; \mathcal{D})$.

**Assumption 2.** *There exist a function* $\Delta : \mathbb{R}^+ \mapsto \mathbb{R}^+$ *such that for any dataset* $S_2 \in (\mathcal{X} \times \mathcal{Y})^{n/2}$ *and* $B > 0$

$$\mathbb{P}_{\mathcal{A}}[\exists (x, y) \in S_2 : |\ell(\mathcal{A}(S_1, B); (x, y))| \geq \Delta(B)] \leq \frac{\min\{\delta, \beta\}}{4K}$$

Specifically, our strategy will be to use the Generalized Exponential Mechanism, $\text{GenExpMech}$, of [RS15] in conjunction with a penalized score function. Roughly, this score function penalty ensures the looser guarantees on the population loss estimate when $B$ is large do not interfere with the loss estimates at smaller values of $B$. We provide the relevant details for $\text{GenExpMech}$ in Appendix C.1. We now state our result.

---

**Algorithm 4** Private Grid Search

---

**Input:** Dataset $\mathcal{S} \in (\mathcal{X} \times \mathcal{Y})^n$, grid parameter $K \in \mathbb{R}$, optimization algorithm: $\mathcal{A} : (\mathcal{X} \times \mathcal{Y})^n \times \mathbb{R} \mapsto \mathbb{R}^d$, privacy parameters $(\epsilon, \delta)$

1: Partition $S$ into two disjoint sets, $S_1$ and $S_2$, of size $\frac{n}{2}$
2: $w_0 = 0$
3: **for** $j \in [K]$ **do**
4:    $B_j = 2^j$
5:    $w_j = \mathcal{A}(S_1, B_j)$
6:    $\widetilde{L}_j = \widehat{L}(w_j; S_2) + \frac{\Delta(B_j) \log(K/\beta)}{n} + \sqrt{\frac{4\|\mathcal{Y}\|^2 \log(K/\beta)}{n}}$
7: **end for**
8: Set $j^*$ as the output of GenExpMech run with privacy parameter $\frac{\epsilon}{2}$, confidence parameter $\frac{\beta}{4}$, and sensitivity/score pairs $(0, \|\mathcal{Y}\|^2), (\Delta(B_1), \widetilde{L}_1)..., (\Delta(B_K), \widetilde{L}_K)$,
9: Output $w_{j^*}$

---

**Theorem 7.** *Let $\ell : \mathbb{R}^d \times (\mathcal{X} \times \mathcal{Y})$ be a smooth non-negative loss function such that $\ell(0, (x, y)) \leq \|\mathcal{Y}\|^2$ for any $x, y \in (\mathcal{X} \times \mathcal{Y})$. Let $\epsilon, \delta, \beta \in [0, 1]$. Let $K > 0$ satisfy $\mathsf{ERR}(2^K) \geq \|\mathcal{Y}\|^2$. Let $\mathcal{A}$ be an $(\frac{\epsilon}{2K}, \frac{\delta}{2K})$-DP algorithm satisfying Assumption 2. Then Algorithm 4 is $(\epsilon, \delta)$-DP. Further, if $\mathcal{A}$ satisfies Assumption 1 and $S_1 \sim \mathcal{D}^{n/2}$ then Algorithm 4 outputs $\bar{w}$ s.t. with probability at least $1 - \beta$,*

$$\varepsilon_{\mathsf{risk}}(\bar{w}; \mathcal{D}) \leq \min \left\{ \|\mathcal{Y}\|^2, \mathsf{ERR}(2 \max \{\|w^*\|, 1\}) + \sqrt{\frac{4\|\mathcal{Y}\|^2 \log(4K/\beta)}{n}} + \frac{5\Delta(2 \max \{\|w^*\|, 1\})}{n\epsilon} \right\}.$$

We note that we develop a generic confidence boosting approach to obtain high probability guarantees from our previously described algorithms in Section 5, and thus obtaining algorithms which satisfy 1 is straightforward. We provide more details on how our algorithms satisfy Assumption 2 in Appendix C.4. The following Theorem details the guarantees implied by this method for output perturbation with boosting (see Theorems 13,15). Full details are in Appendix C.3.

**Theorem 8.** *Let $K, \epsilon, \delta, \beta > 0$ and $\mathcal{A}$ be the algorithm formed by running 3 with boosting and privacy parameters $\epsilon' = \frac{\epsilon}{K}$, $\delta' = \frac{\delta}{K}$. Then there exists a setting of $K$ such that $K = \Theta \left( \log \left( \max \left\{ \frac{\|\mathcal{Y}\|\sqrt{n}}{\|\mathcal{X}\|\sqrt{H}}, \frac{\|\mathcal{Y}\|^2 (n\epsilon)^{2/3}}{\sqrt{H}\|\mathcal{X}\|^2} \right\} \right) \right)$ and Algorithm 4 run with $\mathcal{A}$ and $K$ is $(\epsilon, \delta)$-DP and when given $S \sim \mathcal{D}^n$, satisfies the following w.p. at least $1 - \beta$ (letting $B^* = 2 \max \{\|w^*\|, 1\}$)*

$$\varepsilon_{\mathsf{risk}}(\widehat{w}) = \widetilde{O} \Bigg( \min \Bigg\{ \|\mathcal{Y}\|^2, \frac{\left(\sqrt{H}B^*\|\mathcal{X}\|\right)^{4/3} \|\mathcal{Y}\|^{2/3} + \left(\sqrt{H}B^*\|\mathcal{X}\|\right)^2}{(n\epsilon)^{2/3}}$$

$$+ \frac{\sqrt{H}B^*\|\mathcal{X}\| \max \{\|\mathcal{Y}\|, 1\} + \|\mathcal{Y}\|^2}{\sqrt{n}} + \frac{\|\mathcal{Y}\|^2 + H(B^*\|\mathcal{X}\|)^2}{n\epsilon} \Bigg\} \Bigg).$$

**Confidence Boosting:** We give an algorithm to boost the confidence of unconstrained, smooth DP-SCO (with possibly non-Lipschitz losses). We split the dataset $S$ into $m + 1$ chunks and run an $(\epsilon, \delta)$-DP algorithm over the $m$ chunks to get $m$ models, and then use Report Noisy Max mechanism to select a model with approximately the least empirical risk. We show that this achieves the optimal rate of $\widetilde{O} \left( \frac{\sqrt{H}B\|\mathcal{X}\|\|\mathcal{Y}\|}{\sqrt{n}} \right)$ whereas the previous high probability result of [SST10] had an additional $\widetilde{O} \left( \frac{\|\mathcal{Y}\|^2}{\sqrt{n}} \right)$ term, which was also limited to only GLMs. The key idea is that non-negativity, convexity, smoothness and loss bounded at zero, all together enable strong bounds on the variance of the loss, and consequently give stronger concentration bounds. The details are deferred to Appendix D.

## Acknowledgments and Disclosure of Funding

RA and EU are supported, in part, by NSF BIGDATA award IIS-1838139 and NSF CAREER award IIS-1943251. RB's and MM's work on this research is supported by NSF Award AF-1908281 and Google Faculty Research Award. CG's research is partially supported by INRIA Associate Teams project, FONDECYT 1210362 grant, and National Center for Artificial Intelligence CENIA FB210017, Basal ANID. Part of this work was done while CG was at the University of Twente.

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
