# A   Missing Proofs from Section 3.1 (Smooth GLMs)

## A.1   Utility Lemmas

**Fact 1.** *[SSBD14]  For a $\widetilde{H}$-smooth non-negative function $f$, and for any $u \in \mathrm{dom}(f)$, we have* $\|\nabla f(u)\| \leq \sqrt{4\widetilde{H}f(u)}.$

## A.2   Proof of Lemma 2

From the self-bounding property (Fact 1), the $\|\mathcal{Y}\|^2$ bound on loss at zero, and smoothness, we have the following bound on the gradient:

$$
\begin{aligned}
\|\nabla \ell(w;(x,y))\| &\leq \|\nabla \ell(0;(x,y))\| + \|\nabla \ell(w;(x,y)) - \nabla \ell(0;(x,y))\| \\
&\leq 2\sqrt{H}\,\|x\|\,\ell(0;(x,y)) + H\,\|x\|^2\,\|w\| \\
&\leq 2\left(\sqrt{H}\,\|\mathcal{Y}\|\,\|x\| + H\,\|x\|^2\,\|w\|\right).
\end{aligned}
\tag{2}
$$

**Lemma 3.** *For any $w \in \mathcal{B}_B$ and any $(x,y) \in (\mathcal{X} \times \mathcal{Y})$ it holds that $\ell(w;(x,y)) \leq 3(\|\mathcal{Y}\|^2 + HB^2\|\mathcal{X}\|^2)$.*

*Proof.* Using the fact that the loss function is $G = \|\mathcal{Y}\|\sqrt{H}\|\mathcal{X}\| + BH\|\mathcal{X}\|^2$-Lipschitz in the constraint set (Lemma 2) we have

$$
\begin{aligned}
\ell(w;(x,y)) &\leq \ell(0;(x,y)) + |\ell(w;(x,y)) - \ell(0;(x,y))| \\
&\leq \|\mathcal{Y}\|^2 + G\,\|w\| \\
&\leq \|\mathcal{Y}\|^2 + \|\mathcal{Y}\|\sqrt{H}B\|\mathcal{X}\| + HB^2\|\mathcal{X}\|^2 \\
&\leq 3\left(\|\mathcal{Y}\|^2 + HB^2\|\mathcal{X}\|^2\right)
\end{aligned}
$$

where the last step follow from AM-GM inequality. $\qquad\square$

## A.3   Low Dimension

Before presenting the proof of Theorem 1, we provide formal statements of its Corollaries.

Corollary 1, stated below, gives an upper bound on excess risk of gradient descent in the non-private setting.

**Corollary 1.** *Let $\ell$ be a non-negative convex $\widetilde{H}$ smooth loss function, bounded at zero by $\|\mathcal{Y}\|^2$. Let $n_0 = \frac{\widetilde{H}B^2}{\|\mathcal{Y}\|^2}$, $\mathcal{W} = \mathcal{B}_B$, $T = n$, and $\eta = \min\left(\frac{B}{\sqrt{T}\sqrt{\widetilde{H}}\|\mathcal{Y}\|}, \frac{1}{4\widetilde{H}}\right)$. Given a dataset $S$ of $n \geq n_0$ i.i.d samples from an unknown distribution $\mathcal{D}$, the excess risk of output of Algorithm 1 with $\sigma = 0$ is bounded as,*

$$
\mathbb{E}[L(\widehat{w};\mathcal{D}) - L(w^*;\mathcal{D})] \leq O\left(\frac{\sqrt{\widetilde{H}}B\|\mathcal{Y}\|}{\sqrt{n}}\right).
$$

Corollary 2 below gives an upper bound on excess risk of noisy gradient descent for non-negative smooth GLMs.

**Corollary 2.** *Let $n_0 = \frac{H\|\mathcal{X}\|^2 B^2}{\|\mathcal{Y}\|^2}$, $\eta = \min\left(\frac{B}{\sqrt{T}\max\left(\sqrt{H}\|\mathcal{X}\|\|\mathcal{Y}\|, \sigma\sqrt{d}\right)}, \frac{1}{4H\|\mathcal{X}\|}\right)$, $\mathcal{W} = \mathcal{B}_B$, $\sigma^2 = \frac{8G^2 T \log(1/\delta)}{n^2\epsilon^2}$ and $T = n$. Let $\ell$ be a non-negative convex $H$ smooth GLM, bounded at zero by $\|\mathcal{Y}\|^2$. Algorithm 1 satisfies $(\epsilon, \delta)$-differential privacy. Given a dataset $S \sim \mathcal{D}^n$, $n \geq n_0$, then the excess risk of output of Algorithm 1 is bounded as,*

$$
\mathbb{E}[L(\widehat{w};\mathcal{D}) - L(w^*;\mathcal{D})] = O\left(\frac{\sqrt{H}\|\mathcal{X}\|B\|\mathcal{Y}\|}{\sqrt{n}} + \frac{\left(\sqrt{H}\|\mathcal{X}\|\|\mathcal{Y}\| + H\|\mathcal{X}\|^2 B^2\right)B\sqrt{d\log(1/\delta)}}{n\epsilon}\right).
$$

### A.3.1 Proof of Theorem 1

Since Algorithm 1 uses a projection step, the iterates always lie in the constraint set $\{w : \|w\| \leq B\}$. Hence, the function over this constraint set is $G$-Lipschitz. From the analysis of Noisy (S)GD in [BST14, BFTGT19], we have that the setting of noise variance $\sigma^2$ ensures that the algorithm satisfies $(\epsilon, \delta)$-DP

We now move to the utility part. We start with the decomposition of excess risk as

$$\mathbb{E}[L(\widehat{w}; \mathcal{D}) - L(w^*; \mathcal{D})] = \mathbb{E}[L(\widehat{w}; \mathcal{D}) - \widehat{L}(\widehat{w}; S)] + \mathbb{E}[\widehat{L}(\widehat{w}; S) - \widehat{L}(w^*; S)] \tag{3}$$

The key arguments are as follows: we first bound generalization gap, or on-average stability (first term in the right hand side above), in terms of average argument stability and excess empirical risk (Lemma 4). We then bound average argument stability in terms of average regret (Lemma 5). Finally, in Lemma 6, we provide bounds on excess empirical risk and average regret of gradient descent. Substituting these in the above equation gives the claimed bound. We now fill in the details.

We start with Lemma 4 which gives the following bound on the generalization gap:

$$\mathbb{E}[L(\mathcal{A}(S); \mathcal{D}) - \widehat{L}(\widehat{w}; S)] \leq \frac{\sqrt{\widetilde{H}}B}{\sqrt{n}\,\|\mathcal{Y}\|}\left(\mathbb{E}[\widehat{L}(\widehat{w}; S) - \widehat{L}(w^*; S)]\right) + \frac{2\sqrt{\widetilde{H}n}\|\mathcal{Y}\|}{B}\varepsilon_{\mathsf{av-stab}}(\mathcal{A})^2 + \frac{\sqrt{\widetilde{H}}B\|\mathcal{Y}\|}{\sqrt{n}} \tag{4}$$

Substituting the bound on $\varepsilon_{\mathsf{av-stab}}(\mathcal{A})$ from Lemma 5, the second term becomes,

$$\frac{2\sqrt{\widetilde{H}n}\|\mathcal{Y}\|}{B}\varepsilon_{\mathsf{av-stab}}(\mathcal{A})^2 \leq \frac{16\widetilde{H}^{3/2}\eta^2 T\|\mathcal{Y}\|}{\sqrt{n}B}\frac{1}{n}\sum_{j=1}^{T}\mathbb{E}[\widehat{L}(w_j; S) - \widehat{L}(w^*; S)] + \frac{16\widetilde{H}^{3/2}\eta^2 T\|\mathcal{Y}\|}{\sqrt{n}B}$$

$$\leq \frac{16\sqrt{\widetilde{H}}B}{\sqrt{n}\|\mathcal{Y}\|}\frac{1}{n}\sum_{j=1}^{T}\mathbb{E}[\widehat{L}(w_j; S) - \widehat{L}(w^*; S)] + \frac{16\sqrt{\widetilde{H}}B}{\sqrt{n}\|\mathcal{Y}\|}$$

$$\leq \frac{16}{n}\sum_{j=1}^{T}\mathbb{E}[\widehat{L}(w_j; S) - \widehat{L}(w^*; S)] + \frac{16\sqrt{\widetilde{H}}B}{\sqrt{n}\|\mathcal{Y}\|}.$$

Substituting the above in Eqn. (4) and using the fact that $\frac{\sqrt{\widetilde{H}}B}{\sqrt{n}\|\mathcal{Y}\|} \leq 1$ from the lower bound on $n$, we get:

$$\mathbb{E}[L(\mathcal{A}(S); \mathcal{D}) - \widehat{L}(\widehat{w}; S)] \leq \left(\mathbb{E}[\widehat{L}(\widehat{w}; S) - \widehat{L}(w^*; S)]\right) + \frac{16}{n}\left(\sum_{j=1}^{T}\mathbb{E}[\widehat{L}(w_j; S) - \widehat{L}(w^*; S)]\right)$$

$$+ \frac{17\sqrt{\widetilde{H}}B\|\mathcal{Y}\|}{\sqrt{n}}.$$

From the excess empirical risk guarantee (Lemma 6), the terms excess empirical risk $\mathbb{E}[\widehat{L}(\widehat{w}; S) - \widehat{L}(w^*; S)]$ and average regret $\frac{1}{n}\left(\sum_{j=1}^{T}\mathbb{E}[\widehat{L}(w_j; S) - \widehat{L}(w^*; S)]\right)$ are both bounded by the same quantity. Thus, substituting the above in Eqn. (3) and substituting the bound from 6, we have,

$$\mathbb{E}[L(\widehat{w}; \mathcal{D}) - L(w^*; \mathcal{D})] \leq \frac{18}{n}\sum_{j=1}^{n}\mathbb{E}[\widehat{L}(w_j; S) - \widehat{L}(w^*; S)] + \frac{17\sqrt{\widetilde{H}}B\|\mathcal{Y}\|}{\sqrt{n}}$$

$$\leq O\left(\frac{\sqrt{\widetilde{H}}B\|\mathcal{Y}\|}{\sqrt{n}} + \frac{\sqrt{d}GB\log(1/\delta)}{n\epsilon}\right).$$

Substituting the value of $G$ completes the proof.

**Lemma 4.** *Let $n \geq \frac{\widetilde{H}B^2}{\|\mathcal{Y}\|^2}$. Let $\ell$ be a non-negative $\widetilde{H}$ smooth convex loss function. Let $S$ be a dataset of $n$ i.i.d. samples from an unknown distribution $\mathcal{D}$, and $w^*$ denote the optimal population risk minimizer. The generalization gap of algorithm $\mathcal{A}$ is bounded as,*

$$\mathbb{E}[L(\mathcal{A}(S); \mathcal{D}) - \widehat{L}(\mathcal{A}(S); S)] \leq \frac{\sqrt{\widetilde{H}}B}{\sqrt{n}\,\|\mathcal{Y}\|}\left(\mathbb{E}[\widehat{L}(\mathcal{A}(S); S) - \widehat{L}(w^*; S)]\right) + \frac{2\sqrt{\widetilde{H}n}\|\mathcal{Y}\|}{B}\varepsilon_{\mathsf{av-stab}}(\mathcal{A})^2$$
$$+ \frac{\sqrt{\widetilde{H}}B\|\mathcal{Y}\|}{\sqrt{n}}.$$

*Proof.* Let $\widehat{w} := \mathcal{A}(S)$, $S^{(i)}$ be the dataset where the $i$-th data point is replaced by an i.i.d. point $(x', y')$ and let $\widehat{w}^{(i)}$ be the corresponding output of $\mathcal{A}$.

A standard fact (see [SSBD14]) is that generalization gap is equal to on-average stability:
$$\mathbb{E}[L(\widehat{w}; \mathcal{D}) - \widehat{L}(\widehat{w}; S)] = \mathbb{E}[\ell(\widehat{w}^{(i)}; (x_i, y_i)) - \ell(\widehat{w}; (x_i, y_i)].$$

From smoothness and self-bounding property, we have,
$$\ell(\widehat{w}^{(i)}; (x_i, y_i)) - \ell(\widehat{w}; (x_i, y_i) \leq \|\nabla\ell(\widehat{w}; (x_i, y_i))\|\left\|\widehat{w} - \widehat{w}^{(i)}\right\| + \frac{\widetilde{H}}{2}\left\|\widehat{w} - \widehat{w}^{(i)}\right\|^2$$
$$\leq 2\sqrt{\widetilde{H}\ell((\widehat{w}; (x_i, y_i)))}\left\|\widehat{w} - \widehat{w}^{(i)}\right\| + \frac{\widetilde{H}}{2}\left\|\widehat{w} - \widehat{w}^{(i)}\right\|^2.$$

Taking expectation, using Cauchy-Schwarz inequality and substituting average argument stability, we get,

$$\mathbb{E}\left[\ell(\widehat{w}^{(i)}; (x_i, y_i)) - \ell(\widehat{w}; (x_i, y_i)\right]$$
$$\leq 2\sqrt{\widetilde{H}\mathbb{E}[\ell((\widehat{w}; (x_i, y_i)))]}\sqrt{\mathbb{E}[\|\widehat{w} - \widehat{w}^{(i)}\|^2]} + \frac{\widetilde{H}}{2}\mathbb{E}\left[\left\|\widehat{w} - \widehat{w}^{(i)}\right\|^2\right] \tag{5}$$
$$\leq 2\sqrt{\widetilde{H}\mathbb{E}[\widehat{L}(\widehat{w}; S)]}\varepsilon_{\mathsf{av-stab}}(\mathcal{A}) + \frac{\widetilde{H}\varepsilon_{\mathsf{av-stab}}(\mathcal{A})^2}{2}$$
$$\leq \frac{\sqrt{\widetilde{H}}B\mathbb{E}[\widehat{L}(\widehat{w}; S)]}{\sqrt{n}\,\|\mathcal{Y}\|} + \frac{\sqrt{\widetilde{H}n}\,\|\mathcal{Y}\|\,\varepsilon_{\mathsf{av-stab}}(\mathcal{A})^2}{B} + \frac{\widetilde{H}\varepsilon_{\mathsf{av-stab}}(\mathcal{A})^2}{2}$$
$$\leq \frac{\sqrt{\widetilde{H}}B}{\sqrt{n}\,\|\mathcal{Y}\|}\left(\mathbb{E}[\widehat{L}(\widehat{w}; S) - \widehat{L}(w^*; S)]\right) + \frac{\sqrt{\widetilde{H}n}\,\|\mathcal{Y}\|\,\varepsilon_{\mathsf{av-stab}}(\mathcal{A})^2}{B} + \frac{\widetilde{H}\varepsilon_{\mathsf{av-stab}}(\mathcal{A})^2}{2}$$
$$+ \frac{\sqrt{\widetilde{H}}B}{\sqrt{n}\,\|\mathcal{Y}\|}\mathbb{E}[\widehat{L}(w^*; S)]$$
$$\leq \frac{\sqrt{\widetilde{H}}B}{\sqrt{n}\,\|\mathcal{Y}\|}\left(\mathbb{E}[\widehat{L}(\widehat{w}; S) - \widehat{L}(w^*; S)]\right) + \left(\frac{\sqrt{\widetilde{H}n}\|\mathcal{Y}\|}{B} + \frac{\widetilde{H}}{2}\right)\varepsilon_{\mathsf{av-stab}}(\mathcal{A})^2$$
$$+ \frac{\sqrt{\widetilde{H}}B\|\mathcal{Y}\|}{\sqrt{n}} \tag{6}$$

where the third inequality follows from AM-GM inequality, and last follows since $w^*$ is the optimal solution: $\mathbb{E}[\widehat{L}(w^*; S)] = L(w^*; \mathcal{D}) \leq L(0; \mathcal{D}) \leq \|\mathcal{Y}\|^2$. Finally, using the lower bound on $n$, we have $\frac{\widetilde{H}}{2} \leq \frac{\sqrt{\widetilde{H}n}\|\mathcal{Y}\|}{B}$, substituting which gives the claimed bound. $\qquad\square$

**Lemma 5.** *The average argument stability for noisy GD (Algorithm 1) run for $T$ iterations with step size $\eta \leq \frac{4}{\widetilde{H}}$ is bounded as*
$$\varepsilon_{\mathsf{av-stab}}(\mathcal{A})^2 \leq \frac{8\widetilde{H}\eta^2 T}{n}\frac{1}{n}\sum_{j=1}^{T}\mathbb{E}[\widehat{L}(w_j; S) - \widehat{L}(w^*; S)] + \frac{8\widetilde{H}\eta^2 T\|\mathcal{Y}\|^2}{n}.$$

*Proof.* The uniform argument stability analysis for (Noisy) (S)GD is limited to the Lipschitz setting [HRS15, BFTGT19, BFGT20] and therefore not directly applicable. We therefore need to modify the arguments to give an average stability analysis in smooth (non-Lipschitz) case.

Let $\widehat{w} := \mathcal{A}(S)$, $S^{(i)}$ be the dataset where the $i$-th data point is replaced by an i.i.d. point $(x', y')$ and let $\widehat{w}^{(i)}$ be the corresponding output of $\mathcal{A}$. Moroever, let $w_i$ denote the iterate of noisy SGD on dataset $S$ and similarly $\widetilde{w}_i^{(i)}$ for dataset $S^{(i)}$.

We simply couple the Gaussian noise sampled at each iteration to be equal on both datasets. Using the fact the the updates are non-expansive, we have

$$\left\| w_{t+1} - w'_{t+1} \right\| \leq \| w_t - w'_t \| + \frac{\eta \left( \| \nabla \ell(w_t; (x_i, y_i)) \| + \| \nabla \ell(w'_t; (x', y')) \| \right)}{n}$$
$$\leq \frac{\eta \sum_{j=1}^{t} \left( \| \nabla \ell(w_j; (x_i, y_i)) \| + \| \nabla \ell(w'_j; (x', y')) \| \right)}{n}.$$

From the self-bounding property (Fact 1), $\left( \| \nabla \ell(w_j; (x_i, y_i)) \| \right) \leq 2\sqrt{\widetilde{H} \ell(w_j; (x_i, y_i))}$. Therefore, we get,

$$\mathbb{E}[\| w_t - w'_t \|^2] \leq \frac{4\widetilde{H}\eta^2 T}{n^2} \sum_{j=1}^{t} \left( \mathbb{E}[\ell(w_j; (x_i, y_i)) + \ell(w'_j; (x', y'))] \right)$$
$$= \frac{8\widetilde{H}\eta^2 T}{n^2} \sum_{j=1}^{t} \mathbb{E}[\widehat{L}(w_j; S)].$$

For the average iterate,

$$\mathbb{E}\left[ \left\| \widehat{w} - \widehat{w}^{(i)} \right\|^2 \right] \leq \frac{1}{T^2} T \sum_{t=1}^{T} \mathbb{E}[\| w_t - w'_t \|^2] \leq \frac{8\widetilde{H}\eta^2 T}{n^2} \sum_{j=1}^{T} \mathbb{E}[\widehat{L}(w_j; S)]$$
$$\leq \frac{8\widetilde{H}\eta^2 T}{n} \frac{1}{n} \sum_{j=1}^{T} \mathbb{E}[\widehat{L}(w_j; S) - \widehat{L}(w^*; S)] + \frac{8\widetilde{H}\eta^2 T}{n} \mathbb{E}[\widehat{L}(w^*; S)]$$
$$\leq \frac{8\widetilde{H}\eta^2 T}{n} \frac{1}{n} \sum_{j=1}^{T} \mathbb{E}[\widehat{L}(w_j; S) - \widehat{L}(w^*; S)] + \frac{8\widetilde{H}\eta^2 T \|\mathcal{Y}\|^2}{n}$$

where the last inequality follows since $w^*$ is the population risk minimizer: $\mathbb{E}[\widehat{L}(w^*; S)] = L(w^*; \mathcal{D})] \leq L(0; \mathcal{D}) \leq \|\mathcal{Y}\|^2$. $\qquad \square$

**Lemma 6.** *Let* $n \geq \frac{\widetilde{H}B^2}{\|\mathcal{Y}\|^2}$, $\eta = \min\left( \frac{B}{\sqrt{T} \max\left( \sqrt{\widetilde{H}} \|\mathcal{Y}\|, \sigma\sqrt{d} \right)}, \frac{1}{4\widetilde{H}} \right)$, $\sigma^2 = \frac{8G^2 T \log(1/\delta)}{n^2 \epsilon^2}$ *and* $T = n$.

*We have,*

$$\mathbb{E}\left[ \widehat{L}(\widehat{w}; S) - \widehat{L}(w^*; S) \right] \leq \frac{1}{T} \sum_{j=1}^{T} \mathbb{E}\left[ \widehat{L}(w_j; S) - \widehat{L}(w^*; S) \right] \leq O\left( \frac{\sqrt{\widetilde{H}} B \|\mathcal{Y}\|}{\sqrt{n}} + \frac{\sqrt{d} G \log(1/\delta)}{n\epsilon} \right).$$

*where* $w^*$ *is the (minimum norm) population risk minimizer.*

*Proof.* From standard analysis of (S)GD,

$$\mathbb{E}\left[ \| w_{t+1} - w^* \|^2 \right] \leq \mathbb{E}\left[ \left\| w_t - \eta \left( \nabla \widehat{L}(w_t; S) + \xi_t \right) - w^* \right\|^2 \right]$$
$$\leq \mathbb{E}\left[ \| w_t - w^* \|^2 \right] + \eta^2 \mathbb{E}\left[ \left\| \nabla \widehat{L}(w_t; S) \right\|^2 \right] + \eta^2 \sigma^2 d - 2\eta \mathbb{E}\left[ \widehat{L}(w_t; S) - \widehat{L}(w^*; S) \right]$$
$$\leq \mathbb{E}\left[ \| w_t - w^* \|^2 \right] + 4\eta^2 \widetilde{H} \mathbb{E}[\widehat{L}(w_t; S)] + \eta^2 \sigma^2 d - 2\eta \mathbb{E}\left[ \widehat{L}(w_t; S) - \widehat{L}(w^*; S) \right]$$

where the last inequality follows from self-bounding property (Fact 1). Rearranging, and using the fact that $\mathbb{E}[\widehat{L}(w^*; S) = L(w^*; \mathcal{D})] \leq L(0; \mathcal{D}) \leq \|\mathcal{Y}\|^2$ we get,

$$\left(1 - 2\eta\widetilde{H}\right)\mathbb{E}\left[\widehat{L}(w_t; S) - \widehat{L}(w^*; S)\right] \leq \frac{\mathbb{E}\left[\|w_t - w^*\|^2 - \|w_{t+1} - w^*\|^2\right]}{2\eta} + 2\eta\left(\widetilde{H}\|\mathcal{Y}\|^2 + \sigma^2 d\right)$$

From the choice of $\eta$, we have $\left(1 - 2\eta\widetilde{H}\right) \geq \frac{1}{2}$. Averaging over $T$ iterations, we get that the average regret is,

$$\frac{1}{T}\sum_{j=1}^{T}\mathbb{E}\left[\widehat{L}(w_j; S) - \widehat{L}(w^*; S)\right] \leq \frac{B^2}{\eta T} + 4\eta\left(\widetilde{H}\|\mathcal{Y}\|^2 + \sigma^2 d\right).$$

Setting $\eta = \min\left(\frac{B}{\sqrt{T}\max\left(\sqrt{\widetilde{H}}\|\mathcal{Y}\|, \sigma\sqrt{d}\right)}, \frac{1}{4\widetilde{H}}\right)$, we get

$$\frac{1}{T}\sum_{j=1}^{T}\mathbb{E}\left[\widehat{L}(w_j; S) - \widehat{L}(w^*; S)\right] = O\left(\frac{\sqrt{\widetilde{H}}B\|\mathcal{Y}\|}{\sqrt{T}} + \frac{B\sqrt{d}\sigma}{\sqrt{T}} + \frac{\widetilde{H}B^2}{T}\right).$$

Finally, substituting $\sigma^2 = \frac{8G^2 T \log(1/\delta)}{n^2\epsilon^2}$, $T = n$ and using the lower bound on $n$ gives the claimed bound on average regret. Applying convexity to lower bound average regret by excess empirical risk of $\widehat{w}$ gives the same bound on excess empirical risk. $\qquad\square$

### A.4 High Dimension

*Proof of Theorem 2.* Let $\alpha \leq 1$ be a parameter to be set later. From the JL property with $k = O\left(\frac{\log(2n/\delta)}{\alpha^2}\right)$, with probability at least $1 - \delta/2$, for all data points $x_i$, $\|\Phi x_i\| \leq (1 + \alpha)\|x_i\| \leq 2\|\mathcal{X}\|$, and $\|\Phi w^*\|^2 \leq 2\|w^*\|^2 \leq 2B^2$.

Further, by Lemma 2, for any $w$ in the embedding space with $\|w\|^2 \leq B$, with probability at least $1 - \delta/2$, the loss is $G$ Lipschitz where $G = 2\|\mathcal{Y}\|\sqrt{H}\|\mathcal{X}\| + 2HB\|\mathcal{X}\|^2$. The privacy guarantee now follows from the privacy of Noisy SGD and post-processing.

For the utility guarantee, let $L(w; \Phi\mathcal{D})$ and $\widehat{L}(w; \Phi S)$ denote population and empirical risk (resp) where test and training feature vectors (resp) are mapped using $\Phi$. The excess risk can be decomposed as:

$$\mathbb{E}\left[L(\Phi^\top\widetilde{w}; \mathcal{D}) - L(w^*; \mathcal{D})\right] = \mathbb{E}\left[L(\widetilde{w}; \Phi\mathcal{D}) - L(\Phi w^*; \Phi\mathcal{D})\right] + \mathbb{E}\left[L(\Phi w^*; \Phi\mathcal{D}) - L(w^*; \mathcal{D})\right]. \tag{7}$$

The first term in Eqn. (7) is bounded by the utility guarantee of the DP-SCO method. In particular, from Lemma 7 (below), we have

$$\mathbb{E}[L(\widetilde{w}; \Phi\mathcal{D}) - L(\Phi w^*; \Phi\mathcal{D})] = O\left(\frac{\sqrt{H}\|\mathcal{X}\|B\|\mathcal{Y}\|}{\sqrt{n}} + \frac{\left(\sqrt{H}\|\mathcal{X}\|\|\mathcal{Y}\| + H\|\mathcal{X}\|^2 B^2\right)B\sqrt{k\log(1/\delta)}}{n\epsilon}\right).$$

The second term in Eqn. (7) is bounded by the JL property together with smoothness and the fact that $w^*$ is the optimal solution thus $\nabla L(w^*; \mathcal{D}) = 0$. This gives us

$$\mathbb{E}[L(\Phi w^*; \Phi\mathcal{D})] - L(w^*; \mathcal{D}) \leq \frac{H}{2}\mathbb{E}\left[|\langle\Phi w^*, \Phi x\rangle - \langle w^*, x\rangle|^2\right] \leq \frac{\alpha^2 H \|w^*\|^2 \|\mathcal{X}\|^2}{2} \leq \widetilde{O}\left(\frac{HB^2\|\mathcal{X}\|^2}{k}\right).$$

Combining, we get,

$$\mathbb{E}\left[L(\Phi^\top\widetilde{w}; \mathcal{D}) - L(w^*; \mathcal{D})\right] \leq \widetilde{O}\left(\frac{\sqrt{H}\|\mathcal{X}\|B\|\mathcal{Y}\|}{\sqrt{n}} + \frac{\left(\sqrt{H}\|\mathcal{X}\|\|\mathcal{Y}\| + H\|\mathcal{X}\|^2 B\right)B\sqrt{k\log(1/\delta)}}{n\epsilon} + \frac{HB^2\|\mathcal{X}\|^2}{k}\right).$$

Setting $k = O\left(\frac{BH\|\mathcal{X}\|\log(2n/\delta)n\epsilon}{G}\right)^{2/3}$ completes the proof.

$\square$

**Lemma 7.** *Let $\Phi \in \mathbb{R}^{d \times k}$ be a data-oblivious JL matrix. Let $S = \{(x_i, y_i)\}_{i=1}^n$ of $n$ i.i.d data points and let $\Phi S := \{(\Phi x_i, y_i)\}_{i=1}^n$. Let $\widetilde{w} \in \mathbb{R}^k$ be the average iterate returned by Algorithm 1 with $\sigma^2 = O\left(\frac{G^2\|\mathcal{X}\|^2\log(1/\delta)}{n^2\epsilon^2}\right)$ on dataset $\Phi S$. For $k = \Omega(\log(n))$, the excess risk of $\widetilde{w}$ on $\Phi\mathcal{D}$ is bounded as,*

$$\mathbb{E}[L(\widetilde{w}; \Phi\mathcal{D}) - L(\Phi w^*; \Phi\mathcal{D})] \leq O\left(\frac{\sqrt{H}B\|\mathcal{X}\|\|\mathcal{Y}\|}{\sqrt{n}} + \frac{\left(\sqrt{H}\|\mathcal{X}\|\|\mathcal{Y}\| + H\|\mathcal{X}\|^2B\right)B\sqrt{k\log(1/\delta)}}{n\epsilon}\right).$$

*Proof.* Let $S^{(i)}$ be the dataset where the $i$-th data point is replaced by an i.i.d. point $(x', y')$ and let $\widetilde{w}^{(i)}$ be the corresponding output of Noisy-SGD on $\Phi S^{(i)}$. Define $\overline{S} := \{S, (x', y')\}$ and let $H(\Phi, \overline{S})$ denote an upper bound on the smoothness parameter of the family of loss function $\{\ell(w; (\Phi x; y))\}_{(x,y) \in \overline{S}}$.

We want to apply Theorem 1, but the theorem requires that $n \geq \frac{2H\|\mathcal{X}\|^2B^2}{\|\mathcal{Y}\|^2}$. In the proof below, we will use it to bound $H(\Phi, \overline{S}) \leq \frac{n\|\mathcal{Y}\|^2}{B^2}$. We use the JL property to get this. Let $\alpha \leq 1$ be a parameter to be set later. Note that $H(\Phi, \overline{S}) \leq H \sup_{(x,y) \in \overline{S}} \|\Phi x\|^2 \leq H(1+\alpha)\|\mathcal{X}\|^2 \leq 2H\|\mathcal{X}\|^2$ with probability at least $1 - \delta$ for $k = O\left(\frac{\log(2n/\delta)}{\alpha^2}\right)$. Thus, if we assume $n \geq \frac{2H\|\mathcal{X}\|^2B^2}{\|\mathcal{Y}\|^2}$, then w.h.p. $H(\Phi, \overline{S}) \leq \frac{n\|\mathcal{Y}\|^2}{B^2}$. Also from the JL property, $\|\Phi w^*\| \leq 2B$.

Decomposing excess risk and writing generalization gap as on-average stability, we have
$$\mathbb{E}_{\Phi, S}[L(\widetilde{w}; \Phi\mathcal{D}) - L(\Phi w^*; \Phi\mathcal{D})] = \mathbb{E}_{\Phi, S}[L(\Phi^\top\widetilde{w}; \mathcal{D}) - \widehat{L}(\Phi^\top\widetilde{w}; S)] + \mathbb{E}_{\Phi, S}[\widehat{L}(\Phi^\top\widetilde{w}; S) - \widehat{L}(\Phi w^*; \Phi S)]$$
$$= \mathbb{E}_{\overline{S}, \Phi}[\ell(\widetilde{w}^{(i)}; (\Phi x_i, y_i)) - \ell(\widetilde{w}; (\Phi x_i, y_i)) + \widehat{L}(\Phi^\top\widetilde{w}; S) - \widehat{L}(\Phi w^*; \Phi S)].$$

We now fix the randomness of $\overline{S}$ and bound the terms in high probability w.r.t. the random $\Phi$. Let $\text{IAS}(\overline{S}, i) = \|\widehat{w} - \widehat{w}^{(i)}\|$ denote the instance argument stability. Repeating the analysis in Theorem 1, from Eqn. (5), the first term is bounded as,

$$\ell(\widetilde{w}^{(i)}; (\Phi x_i, y_i)) - \ell(\widetilde{w}; (\Phi x_i, y_i)) \leq \frac{\sqrt{H(\Phi, \overline{S})}B}{\sqrt{n}Y}\left(\ell(\widehat{w}; (\Phi x_i, y_i)) - \widehat{L}(\Phi w^*; \Phi S)\right)$$
$$+ \left(\frac{\sqrt{H(\Phi, \overline{S})n}\|\mathcal{Y}\|}{B} + \frac{H(\Phi, \overline{S})}{2}\right)\text{IAS}(\overline{S}, i)^2 + \frac{\sqrt{H(\Phi, \overline{S})}B\|\mathcal{Y}\|}{\sqrt{n}}$$
$$\leq \frac{2\sqrt{H}\|\mathcal{X}\|B}{\sqrt{n}Y}\left(\ell(\widehat{w}; (\Phi x_i, y_i)) - \widehat{L}(\Phi w^*; \Phi S)\right)$$
$$+ \frac{4\sqrt{Hn}\|\mathcal{X}\|\|\mathcal{Y}\|}{B}\text{IAS}(\overline{S}, i)^2 + \frac{2\sqrt{HB\|\mathcal{Y}\|}\|\mathcal{X}\|}{\sqrt{n}} \quad (8)$$

where the last inequality holds from application of JL property: with probability at least $1 - \delta$, $H(\Phi, \overline{S}) \leq \frac{n\|\mathcal{Y}\|^2}{B^2}$ (from the lower bound on $n$) and $H(\Phi, \overline{S}) \leq 2H\|\mathcal{X}\|^2$.

As in the proof of Lemma 5, $\text{IAS}(\overline{S}, i)$ is bounded as,
$$\text{IAS}(\overline{S}, i)^2$$
$$\leq \frac{8H(\Phi, \overline{S})\eta^2T}{n}\frac{1}{n}\sum_{j=1}^T \mathbb{E}[\ell(w_j; (\Phi x_i, y_i)) + \ell(w'_j; (\Phi x', y')) - 2\widehat{L}(\Phi w^*; \Phi S)] + \frac{8H(\Phi, \overline{S})\eta^2T\|\mathcal{Y}\|^2}{n}$$
$$\leq \frac{16H\|\mathcal{X}\|^2\eta^2T}{n}\frac{1}{n}\sum_{j=1}^T \mathbb{E}[\ell(w_j; (\Phi x_i, y_i)) + \ell(w'_j; (\Phi x', y')) - 2\widehat{L}(\Phi w^*; \Phi S)] + \frac{16H\|\mathcal{X}\|^2\eta^2T\|\mathcal{Y}\|^2}{n}.$$

Substituting the above in equation 8, taking expectation with respect to $\overline{S}$ and from manipulations as in the proof of Theorem 1, we get that with probability at least $1 - \delta$, we have

$$\mathbb{E}_S[L(\Phi^\top \widetilde{w}; \mathcal{D}) - \widehat{L}(\Phi^\top \widetilde{w}; S)] \leq 2\left(\mathbb{E}[\widehat{L}(\widehat{w}; \Phi S) - \widehat{L}(\Phi w^*; \Phi S)]\right)$$
$$+ \frac{32}{n}\left(\sum_{j=1}^{T}\mathbb{E}[\widehat{L}(w_j; \Phi S) - \widehat{L}(w^*; \Phi S)]\right) + \frac{34\sqrt{\widetilde{H}}B\|\mathcal{Y}\|}{\sqrt{n}}.$$

Let $G(\Phi, S) = G = 2\|\mathcal{Y}\|\sqrt{H(\Phi, \overline{S})} + 2H(\Phi, \overline{S})\|\Phi w^*\|$ denote the Lispchitzness parameter of the family of loss functions $\{\ell(w; (\Phi x, y)\}_{(x,y) \in \overline{S}}$. From the analysis in Lemma 6, the average regret and excess empirical risk terms are both bounded by the following quantity with high probability.

$$O\left(\frac{\sqrt{H(\Phi, \overline{S})}B\|\mathcal{Y}\|}{\sqrt{n}} + \frac{\sqrt{k}G(\Phi, S)\log(1/\delta)}{n\epsilon}\right) \leq O\left(\frac{\sqrt{H}B\|\mathcal{Y}\|}{\sqrt{n}} + \frac{\sqrt{k}G\log(1/\delta)}{n\epsilon}\right).$$

This gives the following high probability bound,

$$\mathbb{E}[L(\widetilde{w}; \Phi\mathcal{D}) - L(\Phi w^*; \Phi\mathcal{D})] \leq O\left(\frac{\sqrt{H}B\|\mathcal{X}\|\|\mathcal{Y}\|}{\sqrt{n}} + \frac{\left(\sqrt{H}\|\mathcal{X}\|\|\mathcal{Y}\| + H\|\mathcal{X}\|^2 B\right)B\sqrt{k\log(1/\delta)}}{n\epsilon}\right).$$

For the in-expectation bound, note that in the above proof the JL property was used for: with probability at least $1 - \delta$, $\sup_{(x,y) \in \overline{S}}\|\Phi x\| \leq \left(1 + \frac{\sqrt{\log(n/\delta)}}{k}\right)\|\mathcal{X}\|$ and $\|\Phi w^*\| \leq \left(1 + \frac{\sqrt{\log(n/\delta)}}{k}\right)B$. All these quantities appear in the numerator in the above bound. Therefore the excess risk random variable w.r.t $\Phi$ has a tail with a $\text{poly}\left(\frac{\sqrt{\log(n/\delta)}}{k}\right)$ term. This is a (non-centered) sub-Weibull random variable and from equivalence of tail and moments bounds (e.g. Theorem 3.1 in [VGNA20]) and $\frac{\log(n)}{k} \leq O(1)$, we get the claimed expectation bound.

$\square$

## A.5 Constrained Regularized ERM with Output Perturbation

We here state a key result from [SST10]. Let $\mathfrak{R}_n(B, \|\mathcal{X}\|)$ denote the expected Rademacher complexity of linear predictors with norm bound by $B$, with $n$ datapoints and norm of each point bounded by $\|\mathcal{X}\|$.

**Theorem 9.** *[[SST10]] Let $\ell$ be an $H$-smooth GLM and let $R$ be a bound on the loss function. For a dataset $S$ of $n$ i.i.d samples, with probability at least $1 - \beta$, for all $w$ such that $\|w\| \leq B$, we have*

$$L(w; \mathcal{D}) \leq \widehat{L}(w; S) + O\left(\sqrt{\widehat{L}(w; S)}\left(\sqrt{H}\log^{1.5}(n)\mathfrak{R}_n(B, \|\mathcal{X}\|) + \sqrt{\frac{R\log(1/\beta)}{n}}\right)\right.$$
$$\left. + H\log^3(n)\mathfrak{R}_n^2(B, \|\mathcal{X}\|) + \frac{R\log(1/\beta)}{n}\right). \tag{9}$$

**Corollary 3.** *Let $\widetilde{w} = \arg\min_{w \in \mathcal{B}_B}\left\{\widehat{L}(w; S)\right\}$ and $n \geq \frac{HB^2\|\mathcal{X}\|^2}{\|\mathcal{Y}\|^2}$. Then with probability at least $1 - \beta$ under the randomness of $S$ we have*

$$L(\widetilde{w}; \mathcal{D}) - \widehat{L}(\widetilde{w}; S) = \widetilde{O}\left(\frac{\sqrt{H}B\|\mathcal{X}\|\|\mathcal{Y}\|\sqrt{\log(1/\beta)}}{\sqrt{n}} + \frac{\|\mathcal{Y}\|^2\log(1/\beta)}{\sqrt{n}}\right).$$

*Further, in expectation under the randomness of S it holds that*

$$\mathbb{E}[L(\widetilde{w}; \mathcal{D}) - \widehat{L}(\widetilde{w}; S)] = \widetilde{O}\left(\frac{\sqrt{H}\|\mathcal{Y}\|B\|\mathcal{X}\|}{\sqrt{n}} + \frac{\|\mathcal{Y}\|^2}{\sqrt{n}}\right).$$

*Proof.* In our application, $w$ will be the output of regularized ERM $\widetilde{w}$, thus $\widehat{L}(\widetilde{w}; S) \leq \widehat{L}_\lambda(0; S) \leq \|\mathcal{Y}\|^2$.

From Lemma 3 we have that $R \leq 3(\|\mathcal{Y}\|^2 + HB^2\|\mathcal{X}\|^2)$. Also, $\mathfrak{R}_n(B, \|\mathcal{X}\|) \leq O\left(\frac{B\|\mathcal{X}\|}{\sqrt{n}}\right)$ [SSBD14]. We now plug in the quantities into the above theorem to get that with probability at least $1 - \beta$,

$$
\begin{aligned}
&L(\widetilde{w}; \mathcal{D}) - \widehat{L}(\widetilde{w}; S) \\
&= \widetilde{O}\Bigg( \frac{\sqrt{H}B\|\mathcal{X}\|}{\sqrt{n}} \left( \|\mathcal{Y}\| + \frac{\sqrt{H}B\|\mathcal{X}\|}{\sqrt{n}} \right) \\
&\quad + \frac{\left(\|\mathcal{Y}\| + \sqrt{H}B\|\mathcal{X}\|\right)\sqrt{\log(1/\beta)}}{\sqrt{n}} \left( \|\mathcal{Y}\| + \frac{\left(\|\mathcal{Y}\| + \sqrt{H}B\|\mathcal{X}\|\right)\sqrt{\log(1/\beta)}}{\sqrt{n}} \right) \Bigg) \\
&= \widetilde{O}\left( \frac{\sqrt{H}B\|\mathcal{X}\|\|\mathcal{Y}\|\sqrt{\log(1/\beta)}}{\sqrt{n}} + \frac{\|\mathcal{Y}\|^2\log(1/\beta)}{\sqrt{n}} \right)
\end{aligned}
$$

where the last follows by simplifications using the lower bound on $n$.

For the in expectation result, observe that the above Eqn (9) is a tail bound for a sub-Gamma random variable with variance parameter $O\left(\frac{\sqrt{H}B\|\mathcal{X}\|\|\mathcal{Y}\|}{\sqrt{n}}\right)$ and scale parameter $O\left(\frac{\|\mathcal{Y}\|^2}{\sqrt{n}}\right)$. From the equivalence of tail and moment bounds of sub-Gamma random variables, [BLM13] we get the claimed in-expectation bound. $\qquad\square$

## A.6 Proof of Theorem 3

*Proof.* Recall from Lemma 2 that the (un-regularized) loss is $G$-Lipschitz on the constraint set with $\|w\| \leq B$, where $\widetilde{G} := \left(\sqrt{H}\|\mathcal{Y}\| + HB\|\mathcal{X}\|\right)\|\mathcal{X}\|$. Note that from a standard analysis [BE02], we get that $\ell_2$ sensitivity (or uniform argument stability) is $O\left(\frac{G}{n\lambda}\right)$, hence $\sigma^2 = O\left(\frac{G^2\log(1/\delta)}{\lambda^2n^2\epsilon^2}\right)$ ensures $(\epsilon, \delta)$-DP.

For the utility analysis, the excess risk can be decomposed as follows,

$$\mathbb{E}[L(\widehat{w}; \mathcal{D}) - L(w^*; \mathcal{D})] = \mathbb{E}[L(\widetilde{w}; \mathcal{D}) - \widehat{L}(\widetilde{w}; S)] + \mathbb{E}[\widehat{L}(\widetilde{w}; S) - \widehat{L}(w^*; S)] + \mathbb{E}[L(\widetilde{w} + \xi; \mathcal{D}) - L(\widetilde{w}; \mathcal{D})].$$

The first term is bounded by the Rademacher complexity result (Corollary 3).

The second term is simply bounded by $\frac{\lambda}{2}\|\widetilde{w}\|^2 \leq \frac{\lambda}{2}B^2$ since $\widetilde{w}$ lies in the constraint set. The third term is bounded using smoothness as follows:

$$
\begin{aligned}
\mathbb{E}[L(\widetilde{w} + \xi; \mathcal{D}) - L(\widetilde{w}; \mathcal{D})] &= \mathbb{E}\left[\phi_y\left(\langle\widetilde{w} + \xi, x\rangle\right) - \phi_y\left(\langle\widetilde{w}, x\rangle\right)\right] \\
&\leq \mathbb{E}\left[\phi_y'(\langle\widetilde{w}, x\rangle)\langle\xi, x\rangle + \frac{H}{2}|\langle\xi, x\rangle|^2\right] \\
&\leq \frac{H}{2}\sigma^2\|\mathcal{X}\|^2 = O\left(\frac{HG^2\|\mathcal{X}\|^2\log(1/\delta)}{\lambda^2n^2\epsilon^2}\right)
\end{aligned}
$$

where the last inequality follows since $\mathbb{E}\xi = 0$ and $\langle\xi, x\rangle \sim \mathcal{N}(0, \sigma^2\|x\|^2)$. We now plug in $\lambda = \left(\frac{G\sqrt{H}\|\mathcal{X}\|}{B}\right)^{2/3}\frac{(\log(1/\delta))^{1/3}}{(n\epsilon)^{2/3}}$. and $G = \left(\sqrt{H}\|\mathcal{Y}\| + HB\|\mathcal{X}\|\right)\|\mathcal{X}\|$ to get,

$$\mathbb{E}[L(\widehat{w}; \mathcal{D}) - L(w^*; \mathcal{D})] \leq \widetilde{O}\left(\frac{\sqrt{H}B\|\mathcal{X}\|\|\mathcal{Y}\| + \|\mathcal{Y}\|^2}{\sqrt{n}} + \frac{\left(\sqrt{H}B\|\mathcal{X}\|\right)^{4/3}\|\mathcal{Y}\|^{2/3} + \left(\sqrt{H}B\|\mathcal{X}\|\right)^2}{(n\epsilon)^{2/3}}\right).$$

$\square$

## A.7 Proof of Theorem 4

*Proof.* Define $F(w; S) = \frac{1}{n} \sum_{(x,y) \in S} (\langle w, x \rangle - y)^2$. Let $d' < \min\{n, d\}$ and $b, p \in [0, 1]$ be parameters to be chosen later. For any $\boldsymbol{\sigma} \in \{\pm 1\}^{d'}$, define the dataset $S_{\boldsymbol{\sigma}}$ which consists of the union of $d'$ subdatasets, $S_1, ..., S_{d'}$ given as follows. Set $\frac{pn}{d'}$ of the feature vectors in $S_j$ as $\|\mathcal{X}\| e_j$ (the rescaled $j$'th standard basis vector) and the rest as the zero vector. Set $\frac{pn}{2d}(1 + b)$ of the labels as $\boldsymbol{\sigma}_j \|\mathcal{Y}\|$ and $\frac{pn}{2d}(1 - b)$ labels as $-\boldsymbol{\sigma}_j \|\mathcal{Y}\|$. Let $w^{\boldsymbol{\sigma}} = \arg\min_{w \in \mathbb{R}^d} \{F(w; S_{\boldsymbol{\sigma}})\}$ be the ERM minimizer of $F(\cdot; S_{\boldsymbol{\sigma}})$. Following from Lemma 2 of [Sha15] we have that for any $\bar{w} \in \mathbb{R}^d$ that

$$F(\bar{w}; S_{\boldsymbol{\sigma}}) - F(w^{\boldsymbol{\sigma}}; S_{\boldsymbol{\sigma}}) \geq \frac{p \|\mathcal{X}\|^2}{2d'} \sum_{j=1}^{d'} (\bar{w}_j - w_j^{\boldsymbol{\sigma}})^2. \tag{10}$$

We will now show lower bounds on the per-coordinate error. Consider any $\boldsymbol{\sigma}$ and $\boldsymbol{\sigma}'$ which differ only at index $j$ for some $j \in [d']$. Note that the datasets $S_{\boldsymbol{\sigma}}$ and $S_{\boldsymbol{\sigma}'}$ differ in $\Delta = \frac{pn}{2d'}[(1 + b) - (1 - b)] = \frac{pbn}{d'}$ points. Let $\tau = w_j^{\boldsymbol{\sigma}} = \frac{\|\mathcal{Y}\| b}{\|\mathcal{X}\|}$ and $\tau' = w_j^{\boldsymbol{\sigma}'} = -\frac{\|\mathcal{Y}\| b}{\|\mathcal{X}\|}$ (i.e. the $j$ components of the empirical minimizers for $S$ and $S_j'$ respectively). Note that $|w_j^{\boldsymbol{\sigma}} - w_j^{\boldsymbol{\sigma}'}| = \frac{2\|\mathcal{Y}\| b}{\|\mathcal{X}\|}$. We thus have by Lemma 1 that for a certain $b = b(\epsilon, n, d, B, p, \|\mathcal{Y}\|)$, $\mathcal{A}$ must satisfy

$$\mathbb{E}\left[|\mathcal{A}(S_{\boldsymbol{\sigma}})_j - w_j^{\boldsymbol{\sigma}}| + |\mathcal{A}(S_{\boldsymbol{\sigma}'})_j - w_j^{\boldsymbol{\sigma}'}|\right] \geq \frac{1}{4} \frac{\|\mathcal{Y}\| b}{\|\mathcal{X}\|}. \tag{11}$$

Since we need $\Delta \leq \frac{1}{\epsilon}$, we must set $b \leq \frac{d'}{pn\epsilon}$. Furthermore, if we are interested in problems with minimizer norm at most $B$, we need $b \leq \frac{\|\mathcal{X}\| B}{\|\mathcal{Y}\| \sqrt{d'}}$ to ensure the norm of the minimizer is bounded by $B$. Balancing these two restrictions, we set $d' = \left(\frac{p \|\mathcal{X}\| Bn\epsilon}{\|\mathcal{Y}\|}\right)^{2/3}$ which yields $b = \left(\frac{\|\mathcal{X}\| B}{\|\mathcal{Y}\| \sqrt{pn\epsilon}}\right)^{2/3}$. Assuming such settings of $b$ and $d'$ are possible (e.g. $b \in [0, 1]$) we can apply Jensen's inequality and the fact that $(a + b)^2 \leq 2(a^2 + b^2)$ to Eqn. (11) to obtain

$$\mathbb{E}\left[|\mathcal{A}(S_{\boldsymbol{\sigma}})_j - w_j^{\boldsymbol{\sigma}}|^2 + |\mathcal{A}(S_{\boldsymbol{\sigma}'})_j - w_j^{\boldsymbol{\sigma}'}|^2\right] \geq \frac{B^{4/3} \|\mathcal{Y}\|^{2/3}}{32(\|\mathcal{X}\| pn\epsilon)^{2/3}}.$$

We will now show this implies there exists a $\boldsymbol{\sigma} \in \{\pm 1\}^{d'}$ such that $F(\mathcal{A}(S_{\boldsymbol{\sigma}})) - F(w^{\boldsymbol{\sigma}}) = \Omega\left(\frac{(\|\mathcal{X}\| B)^{4/3} \|\mathcal{Y}\|^{2/3} p^{1/3}}{(n\epsilon)^{2/3}}\right)$. To prove this, we have the following analysis. Let $U = \{\pm 1\}^{d'}$ and let $\boldsymbol{\sigma}_{-j}$ denote the vector $\boldsymbol{\sigma}$ with its $j$'th component negated. We have

$$\sup_{\boldsymbol{\sigma} \in U} \left\{\mathbb{E}\left[F(\mathcal{A}(S_{\boldsymbol{\sigma}})) - F(w^{\boldsymbol{\sigma}})\right]\right\} \geq \frac{1}{|U|} \sum_{\boldsymbol{\sigma} \in U} \mathbb{E}\left[F(\mathcal{A}(S_{\boldsymbol{\sigma}})) - F(w^{\boldsymbol{\sigma}})\right]$$

$$= \frac{p \|\mathcal{X}\|^2}{d' |U|} \sum_{j \in [d']} \sum_{\boldsymbol{\sigma} \in U} \mathbb{E}\left[|\mathcal{A}(S_{\boldsymbol{\sigma}})_j - w_j^{\boldsymbol{\sigma}}|^2\right]$$

$$= \frac{p \|\mathcal{X}\|^2}{d' |U|} \sum_{j \in [d']} \sum_{\boldsymbol{\sigma} \in U : \boldsymbol{\sigma}_j = 1} \mathbb{E}\left[|\mathcal{A}(S_{\boldsymbol{\sigma}})_j - w_j^{\boldsymbol{\sigma}}|^2 + |\mathcal{A}(S_{\boldsymbol{\sigma}_{-j}})_j - w_j^{\boldsymbol{\sigma}_{-j}}|^2\right]$$

$$\geq \frac{(\|\mathcal{X}\| B)^{4/3} \|\mathcal{Y}\|^{2/3} p^{1/3}}{128 (n\epsilon)^{2/3}}.$$

We recall this bound holds providing the settings of $d'$ and $b$ fall into the range $[1, \min\{n, d\}]$ and $[0, 1]$ respectively. First note $b > 0$ always. Furthermore, $d' < \min\{n, d\}$ and $b < 1$ whenever

$$\frac{\|\mathcal{X}\|^2 B^2}{\|\mathcal{Y}\|^2 n\epsilon} \leq p \leq \min\left\{1, \frac{d^{3/2} \|\mathcal{Y}\|}{\|\mathcal{X}\| Bn\epsilon}\right\}. \tag{12}$$

In the following, assume $B \leq \frac{\|\mathcal{Y}\| \min\{\sqrt{n\epsilon}, \sqrt{d}\}}{\|\mathcal{X}\|}$. Note this is no loss of generality as we can obtain problem instances with arbitrarily large $B$ by adding a dummy point with $x = c' e_{d'+1}$ and

$y = \|\mathcal{Y}\|$ where $c'$ is arbitrarily small so that the minimizer norm is $B$. Under this assumption on $B$, using the restrictions on $p$, it can be verified that $d' > 1$ whenever $\epsilon > \frac{1}{n}$. Thus we have $b \in [0,1]$ and $d' \in [1, \min\{n, d\}]$ as required. Furthermore this assumption on $B$ implies $\frac{\|\mathcal{X}\|^2 B^2}{\|\mathcal{Y}\|^2 n \epsilon} \le \min\left\{1, \frac{d^{3/2}\|\mathcal{Y}\|}{\|\mathcal{X}\|Bn\epsilon}\right\}$ and thus a valid setting of $p$ is possible. We now turn to setting $p$ in a way which satisfies (12). We consider two cases, the high and low dimensional regimes.

**Case 1:** $d \ge \left(\frac{B\|\mathcal{X}\|n\epsilon}{\|\mathcal{Y}\|}\right)^{2/3}$. Setting $p = 1$ gives a lower bound of $\Omega\left(\min\left\{\|\mathcal{Y}\|^2, \frac{B^{4/3}\|\mathcal{X}\|^{4/3}\|\mathcal{Y}\|^{2/3}}{32(n\epsilon)^{2/3}}\right\}\right)$, where the min with $\|\mathcal{Y}\|^2$ from the upper bound on $B$.

**Case 2:** $d \le \left(\frac{B\|\mathcal{X}\|n\epsilon}{\|\mathcal{Y}\|}\right)^{2/3}$. Setting $p = \frac{d^{3/2}\|\mathcal{Y}\|}{B\|\mathcal{X}\|n\epsilon}$ we obtain a bound of $\Omega\left(\min\left\{\|\mathcal{Y}\|^2, \frac{\sqrt{d}B\|\mathcal{X}\|\|\mathcal{Y}\|}{n\epsilon}\right\}\right)$ which we note is no larger than the bound from Case 1 in the low dimensional regime and no smaller than the bound from Case 1 in the high dimensional regime. Thus we can write the total bound as the minimum of these two bounds. The $\|\mathcal{Y}\|^2$ term again comes from the restriction on $B$.

To obtain results for arbitrary $H$, we can set $F(w; S) = \frac{H}{2n}\sum_{(x,y)\in S}(\langle w, x\rangle - \frac{2}{\sqrt{H}}y)^2$. This satisfies $H$-smoothness and loss bounded at zero by $\|\mathcal{Y}\|^2$. Then substituting $\|\mathcal{Y}\|$ in the previous expressions for $2\|\mathcal{Y}\|/\sqrt{H}$ and multiplying through by $H/2$ one obtains the claimed bound. $\qquad\square$

# B  Missing Proofs from Section 4 (Lipschitz GLMs)

## B.1  Proof of Theorem 5

*Proof.* Note that from a standard analysis [BE02], we get that $\ell_2$ sensitivity of the regularized minimizer $\widetilde{w}$ is $\left(\frac{2G\|\mathcal{X}\|}{n\lambda}\right)$, hence $\sigma^2 = \frac{4G^2\|\mathcal{X}\|^2 \log(1/\delta)}{\lambda^2 n^2 \epsilon^2}$ ensures $(\epsilon, \delta)$-DP.

For the utility analysis, the excess risk can be decomposed as follows,

$$\mathbb{E}[L(\widehat{w}; \mathcal{D}) - L(w^*; \mathcal{D})] = \mathbb{E}[L(\widetilde{w}; \mathcal{D}) - \widehat{L}(\widetilde{w}; S)] + \mathbb{E}[\widehat{L}(\widetilde{w}; S) - \widehat{L}(w^*; S)] + \mathbb{E}[L(\widetilde{w} + \xi; \mathcal{D}) - L(\widetilde{w}; \mathcal{D})]$$

The first term is bounded as $O\left(\frac{BG\|\mathcal{X}\|}{\sqrt{n}}\right)$ from Rademacher complexity results on bounded linear predictors [SSBD14].

The second term is simply bounded by $\frac{\lambda}{2}\|\widetilde{w}\|^2 \le \frac{\lambda}{2}B^2$ since $\widetilde{w}$ is the regularized ERM. The third term is bounded via the following

$$\begin{aligned}
\mathbb{E}[L(\widetilde{w} + \xi; \mathcal{D}) - L(\widetilde{w}; \mathcal{D})] &= \mathbb{E}\left[\phi_y\left(\langle \widetilde{w} + \xi, x\rangle\right) - \phi_y\left(\langle\widetilde{w}, x\rangle\right)\right] \\
&\le \mathbb{E}\left[G|\langle\xi, x\rangle|\right] \\
&\le G\sigma\|\mathcal{X}\| \le \frac{4G^2\|\mathcal{X}\|^2\sqrt{\log(1/\delta)}}{\lambda n\epsilon}
\end{aligned}$$

where the last inequality follows since $\mathbb{E}\xi = 0$ and $\langle\xi, x\rangle \sim \mathcal{N}(0, \sigma^2\|x\|^2)$. Now plugging in $\lambda = \frac{G\|\mathcal{X}\|(\log(1/\delta))^{1/4}}{B\sqrt{n\epsilon}}$ obtains the following result

$$\mathbb{E}[L(\widehat{w}; \mathcal{D}) - L(w^*; \mathcal{D})] = O\left(\frac{BG\|\mathcal{X}\|}{\sqrt{n}} + \frac{BG\|\mathcal{X}\|\log(1/\delta)^{1/4}}{\sqrt{n\epsilon}}\right).$$

$\square$

## B.2  Upper Bound using JL Method

**Theorem 10.** *Let* $k = O\left(\log(2n/\delta)n\epsilon\right), \sigma^2 = \frac{8TG^2\|\mathcal{X}\|^2\log(2/\delta)}{n^2\epsilon^2}, \eta = \frac{B}{G\|\mathcal{X}\|\left(1 + \frac{\sqrt{k\log(2/\delta)}}{n\epsilon}\right)T^{3/4}}$*

*and* $T = n^2$. *Algorithm 2 satisfies* $(\epsilon, \delta)$-*differential privacy. Given a dataset* $S$ *of* $n$ *i.i.d samples,*

*the excess risk of its output $\widetilde{w}$ is bounded as*

$$\mathbb{E}[\varepsilon_{\mathsf{risk}}(\widehat{w})] = \widetilde{O}\left(\frac{GB\,\|\mathcal{X}\|}{\sqrt{n}} + \frac{GB\|\mathcal{X}\|}{\sqrt{n}\epsilon}\right).$$

*Proof.* Let $\alpha \leq 1$ be a parameter to be set later. From the JL property, with $k = O\left(\frac{\log(2n/\delta)}{\alpha^2}\right)$, with probability at least $1 - \frac{\delta}{2}$, for feature vectors have $\|\Phi x_i\| \leq (1+\alpha)\|x_i\| \leq 2\|\mathcal{X}\|$, and $\|\Phi w^*\|^2 \leq 2\|w^*\|^2 \leq 2B^2$. Thus, gradient of loss for data point $(x,y)$ in $S$ at any $w$ is bounded as $\|\ell(w;(\Phi x, y)\| = \phi_y'(\langle w, \Phi x\rangle)\|\Phi x\| \leq 2G\|\mathcal{X}\|$. The privacy guarantee thus follows from the privacy of DP-SCO and post-processing.

For the utility guarantee, we decompose excess risk as:

$$\mathbb{E}\left[L(\widehat{w};\mathcal{D}) - L(w^*;\mathcal{D})\right] = \mathbb{E}\left[L(\widetilde{w};\Phi\mathcal{D}) - L(\Phi w^*;\Phi S)\right] + \mathbb{E}\left[L(\Phi w^*;\Phi S) - L(w^*;\mathcal{D})\right]. \quad (13)$$

The first term in Eqn. (13) is bounded by the utility guarantee of the DP-SCO method. In particular, from Lemma 8 (below), we have

$$\mathbb{E}\left[L(\widetilde{w};\Phi\mathcal{D}) - L(\Phi w^*;\Phi S)\right] \leq \frac{GB\,\|\mathcal{X}\|}{\sqrt{n}} + O\left(\frac{G\,\|\mathcal{X}\|\,B\sqrt{k}}{n\epsilon}\right).$$

For the second term in Eqn (13), we use JL-property and Lipschitzness of GLM:

$$\begin{aligned}
\mathbb{E}\left[L(\Phi w^*;\Phi S) - L(w^*;\mathcal{D})\right] &\leq G\mathbb{E}\left[|\langle \Phi x, \Phi w^*\rangle - \langle x, w^*\rangle|\right] \\
&\leq \alpha G\,\|\mathcal{X}\|\,\|w^*\| \\
&= O\left(\frac{G\,\|\mathcal{X}\|\,B\sqrt{\log(2n/\delta)}}{\sqrt{k}}\right).
\end{aligned}$$

Combining, we get,

$$\mathbb{E}\left[L(\widehat{w};\mathcal{D}) - L(w^*;\mathcal{D})\right] \leq \widetilde{O}\left(\frac{G\,\|\mathcal{X}\|\,B}{\sqrt{n}} + \frac{G\,\|\mathcal{X}\|\,B\sqrt{k}}{n\epsilon} + \frac{G\,\|\mathcal{X}\|\,B}{\sqrt{k}}\right).$$

Balancing parameters by setting $k = \widetilde{O}(n\epsilon)$ gives the claimed bound. $\qquad\square$

**Lemma 8.** *Let $\Phi \in \mathbb{R}^{d\times k}$ be a data-oblivious JL matrix. Let $S = \{(x_i, y_i)\}_{i=1}^n$ of $n$ i.i.d data points and let $\Phi S := \{(\Phi x_i, y_i)\}_{i=1}^n$. Let $\widetilde{w} \in \mathbb{R}^k$ be the average iterate returned noisy SGD procedure with Gaussian noise variance $\sigma^2 = O\left(\frac{G^2\|\mathcal{X}\|^2\log(1/\delta)}{n^2\epsilon^2}\right)$ on dataset $\Phi S$. For $k = \Omega\left(\log(n)\right)$, the excess risk of $\widetilde{w}$ on $\Phi\mathcal{D}$ is bounded as,*

$$\mathbb{E}_{\Phi,S}[L(\widetilde{w};\Phi\mathcal{D}) - L(\Phi w^*;\Phi\mathcal{D})] \leq O\left(\frac{GB\|\mathcal{X}\|}{\sqrt{n}} + \frac{GB\|\mathcal{X}\|\sqrt{k\log(1/\delta)}}{n\epsilon}\right). \quad (14)$$

*Proof.* The proof uses the analysis for excess risk bound for Noisy SGD in [BFGT20] with the JL transform. Let $S^{(i)}$ be the dataset where the $i$-th data point is replaced by an i.i.d. point $(x', y')$ and let $\widetilde{w}^{(i)}$ be the corresponding output of Noisy-SGD on $\Phi S^{(i)}$. Define $\overline{S} := \{S, (x', y')\}$ and let $G(\Phi, \overline{S})$ denote an upper bound Lipschitzness parameter of the family of loss functions $\{\ell(w;(\Phi x; y))\}_{(x,y)\in\overline{S}}$.

We decompose the excess risk as:

$$\mathbb{E}_{\Phi,S}[L(\widetilde{w};\Phi\mathcal{D}) - L(\Phi w^*;\Phi\mathcal{D})] = \mathbb{E}_{\Phi,S}[L(\Phi^\top\widetilde{w};\mathcal{D}) - \widehat{L}(\Phi^\top\widetilde{w};S)] + \mathbb{E}_{\Phi,S}[\widehat{L}(\Phi^\top\widetilde{w};S) - \widehat{L}(\Phi w^*;\Phi S)].$$

A well-known fact (see [SSBD14]) is that generalization gap is equal to on-average-stability:

$$\mathbb{E}_{\Phi,S}[L(\Phi^\top \widetilde{w}; \mathcal{D}) - \widehat{L}(\Phi^\top \widetilde{w}; S)] = \mathbb{E}_{\overline{S},\Phi}[\ell(\Phi^\top \widetilde{w}^{(i)}; (x_i, y_i)) - \ell(\Phi^\top \widetilde{w}; (x_i, y_i))]$$
$$= \mathbb{E}_{\overline{S},\Phi}[\ell(\widetilde{w}^{(i)}; (\Phi x_i, y_i)) - \ell(\widetilde{w}; (\Phi x_i, y_i))].$$

From the analysis in Theorem 3.3 from [BFGT20], it follows that

$$\ell(\widetilde{w}^{(i)}; (\Phi x_i, y_i)) - \ell(\widetilde{w}; (\Phi x_i, y_i)) \le G(\Phi; \overline{S}) \left\| \widetilde{w}^{(i)} - \widetilde{w} \right\| \le O\left( G(\Phi; \overline{S})^2 \left( \eta \sqrt{T} + \frac{\eta T}{n} \right) \right). \tag{15}$$

where the last equality follows from the GLM structure of the loss function.

From analysis of Noisy-SGD [BST14], the other term (excess empirical risk) can be bounded as

$$\widehat{L}(\widetilde{w}; \Phi S)] - \widehat{L}(\Phi w^*; \Phi S) \le O\left( \eta \left( G(\Phi; \overline{S})^2 + \frac{k G(\Phi; \overline{S})^2 \log(1/\delta)}{n^2 \epsilon^2} \right) + \frac{\|\Phi w^*\|^2}{T\eta} \right) \tag{16}$$

We now take expectation with respect to $\Phi$. Note that $G(\Phi, \overline{S}) = \sup_{(x,y)\in\overline{S}} \sup_w |g_x(\langle w, \Phi x\rangle)| \|\Phi x\|$, where $g_x$ is the an element of the sub-differential of $\phi_x$ at $\langle w, x\rangle$. By the Lipschitzness assumption $|g_x(\langle w, \Phi x\rangle)| \le G$ and from the JL property, with probability at least $1 - \delta$, $\sup_{(x,y)\in\overline{S}} \|\Phi x\| \le \left( 1 + \frac{\sqrt{\log(n/\delta)}}{k} \right) \|\mathcal{X}\|$. Similarly, $\|\Phi^\top w^*\| \le \left( 1 + \frac{\sqrt{\log(n/\delta)}}{k} \right) B$. Observe that the tail is that of a (non-centered) sub-Gaussian random variable with variance parameter $O\left( \frac{1}{k} \right)$. Using the equivalence of tail and moment bounds of sub-Gaussian random variables [BLM13] and the fact that $\frac{\log(n)}{k} \le O(1)$, we get the following bounds for Eqn. (15) and (16):

$$\mathbb{E}_\Phi \left[ \ell(\widetilde{w}^{(i)}; (\Phi x_i, y_i)) - \ell(\widetilde{w}; (\Phi x_i, y_i)) \right] \le O\left( G^2 \|\mathcal{X}\|^2 \left( \eta \sqrt{T} + \frac{\eta T}{n} \right) \right)$$

$$\mathbb{E}_\Phi \left[ \widehat{L}(\widetilde{w}; \Phi S)] - \widehat{L}(\Phi w^*; \Phi S) \right] \le \widetilde{O}\left( \eta \left( G^2 \|\mathcal{X}\|^2 + \frac{k G^2 \|\mathcal{X}\|^2 \log(1/\delta)}{n^2 \epsilon^2} \right) + \frac{B^2}{T\eta} \right).$$

Finally, as in [BFGT20], taking expectation with respect $\overline{S}$ in the two inequalities above, setting $\eta = \frac{B}{G\|\mathcal{X}\|\left( 1 + \frac{\sqrt{k \log(1/\delta)}}{n\epsilon} \right) T^{3/4}}$ and $T = n^2$ and combining gives the claimed bound. $\qquad\square$

## B.3   Proof of Theorem 6

The proof follows from the more general Theorem 11 stated below. Instantiating Theorem 11 with $p = q = 2$ satisfies all the requirements of our Theorem 6. Finally, let $w^*$ be the population risk minimizer of the hard instance in Theorem 11. We then have,

$$\mathbb{E}[L(\mathcal{A}(S); \mathcal{D}) - \min_w L(w; \mathcal{D})] = \mathbb{E}[L(\mathcal{A}(S); \mathcal{D}) - L(w^*; \mathcal{D})]$$
$$\ge \mathbb{E}[L(\mathcal{A}(S); \mathcal{D}) - \min_{w:\|w\|_2\le B} L(w; \mathcal{D})]$$
$$\ge \mathbb{E}[L(\widehat{w}; \mathcal{D}) - L(\widetilde{w}; \mathcal{D})]$$
$$= \Omega\left( GB\|\mathcal{X}\| \min\left( 1, \frac{1}{\sqrt{n}\epsilon}, \frac{\sqrt{\text{rank}}}{n\epsilon} \right) \right).$$

This completes the proof.

## B.4 Lower bound for Non-Euclidean DP-GLM

**Theorem 11.** *Let $G, \|\mathcal{Y}\|, \|\mathcal{X}\|, B > 0$, $\epsilon \leq 1.2, \delta \leq \epsilon$ and $p, q \geq 1$. For any $(\epsilon, \delta)$-DP algorithm $\mathcal{A}$, there exists sets $\mathcal{X}$ and $\mathcal{Y}$ such that for any $x \in \mathcal{X}$, $\|x\|_q \leq 1$, a distribution $\mathcal{D}$ over $\mathcal{X} \times \mathcal{Y}$, a $G$-Lipschitz GLM loss bounded at zero by $\|\mathcal{Y}\|$ and a $\widetilde{w}$ with $\|\widetilde{w}\|_p \leq B$ such that the output of $\mathcal{A}$ on $S \sim \mathcal{D}^n$ satisfies*

$$\mathbb{E}[L(\mathcal{A}(S); \mathcal{D}) - L(\widetilde{w}; \mathcal{D})] = \Omega\left(GB\|\mathcal{X}\|\min\left(1, \frac{1}{(n\epsilon)^{1/p}}, \frac{(rank)^{(p-1)/p}}{n\epsilon}\right)\right).$$

*Proof.* The construction below is from [SSTT20] with some changes. We provide the complete proof below. Consider the following 1-Lipschitz loss function:

$$\ell(w; (x, y)) = |y - \langle w, x\rangle|.$$

Firstly, we argue that we can assume $\|\mathcal{Y}\| = \infty$. This is because for any arbitrarily large value of $y$ we can translate the above function below so that the the loss is bounded by $\|\mathcal{Y}\|$. However, this translation doesn't change the excess risk. Also, as in [BST14], it suffices to consider $G = 1$, since we can simply scale the 1-Lipschitz loss function and get a factor of the $G$ in the lower bound. Let $d' \leq \min(rank, n\epsilon)$, $0 \leq \alpha \leq 1$ and $\beta > 0$ be parameters to be set later. Since $d' \leq rank$, without loss of generality, we will represent the features $x$ as $d'$ dimensional vectors.

Consider a distribution $\mathcal{D}$, where $x = e_0 := \vec{0}$ with probability $1 - \alpha$, and with probability $\alpha$, $x \sim \text{Unif}(\{\|\mathcal{X}\| e_i\}_{i=1}^{d'})$. Note that $\|x\|_q \leq \|\mathcal{X}\|$ and for any $q \geq 1$ for $x \in \text{supp}(\mathcal{D}_x)$ where $\mathcal{D}_x$ denotes the marginal of $\mathcal{D}$ w.r.t. the $x$ variable. This implies the loss $w \mapsto \ell(w; (x, y))$ is $\|\mathcal{X}\|$-Lipschitz in $\ell_q$-norm, when $x \in \text{supp}(\mathcal{D}_x)$ and for all $y$. Let the fingerprinting code $z \in \{0, 1\}^{d'}$ be drawn from a product distribution with mean $\mu \in [0, 1]^{d'}$ where each co-ordinate $\mu_i \sim \text{Beta}(\beta, \beta)$. Finally we have $y = \frac{B}{(d')^{1/p}}\langle x, z\rangle$. Let $S = \{(x_i, y_i)\}_{i=1}^n$ be $n$ i.i.d. samples drawn from $\mathcal{D}$. Define $\widetilde{w} = \frac{B}{(d')^{1/p}}\mu$; note that $\|\widetilde{w}\|_p \leq B$.

Let $\mathcal{A}$ be any $(\epsilon, \delta)$-DP algorithm, which given $S$ outputs $\widehat{w}$. Its excess risk with respect to $\widetilde{w}$ can be lower bounded as,

$$\mathbb{E}[L(\widehat{w}; \mathcal{D}) - L(\widetilde{w}; \mathcal{D})] = \mathbb{E}[|y - \langle\widehat{w}, x\rangle| - |y - \langle\widetilde{w}, x\rangle|]$$
$$\geq \mathbb{E}[|\langle\widehat{w} - \widetilde{w}, x\rangle| - 2|y - \langle\widetilde{w}, x\rangle|]. \tag{17}$$

The last term above is upper bounded as:

$$\mathbb{E}[|y - \langle\widetilde{w}, x\rangle|] = \frac{B}{(d')^{1/p}}\mathbb{E}[|\langle z, x\rangle - \langle\mu, x\rangle|] = \frac{B\|\mathcal{X}\|}{(d')^{1/p}}\frac{\alpha}{d'}\sum_{i=1}^{d'}\mathbb{E}|z_i - \mu_i|.$$

By direct computation,

$$\mathbb{E}|z_i - \mu_i| = \mathbb{E}[|1 - \mu_i|\mu_i + |-\mu_i|(1 - \mu_i)] = 2\mathbb{E}\mu_i(1 - \mu_i) = \frac{2\beta}{1 + 2\beta}.$$

This gives us that

$$\mathbb{E}[|y - \langle\widetilde{w}, x\rangle|] \leq \frac{2\alpha\beta B\|\mathcal{X}\|}{(1 + 2\beta)(d')^{(p+1)/p}}. \tag{18}$$

The first term in the right hand side Inequality (17) is,

$$\mathbb{E}[|\langle\widehat{w} - \widetilde{\mu}, x\rangle| = \frac{B}{(d')^{1/p}}\mathbb{E}\left|\left\langle\frac{(d')^{1/p}}{B}\widehat{w} - \mu, x\right\rangle\right| = \frac{B\|\mathcal{X}\|\alpha}{(d')^{(p+1)/p}}\mathbb{E}\sum_{j=1}^{d'}\left|\frac{(d')^{1/p}}{B}\widehat{w}_j - \mu_j\right|. \tag{19}$$

Define $v \in \mathbb{R}^{d'}$ with $v_i = \frac{(d')^{1/p}\widehat{w}_i}{B}$. Note that,

$$\sum_{j=1}^{d'}\left|\frac{(d')^{1/p}}{B}\widehat{w}_j - \mu_j\right| = \|v - \mu\|_1 \geq \|v - \mu\|_2 \geq \|\widehat{v} - \mu\|_2 \geq \|\widehat{v} - \mu\|_2^2$$

where the first inequality above follows from relationship between $\ell_1$ and $\ell_2$ norm, the second follows from projection property and $\widehat{v}$ denotes the projection of $v$ onto $[0, 1]^d$, and the last inequality follows from boundedness of coordinates of $\widehat{v} - \mu$.

The key step now is application of the fingerprinting lemma (simplified below, see Lemma B.1 in [SSTT20] for complete statement) from [SU17] which roughly speaking, "relates the error to correlation"; for any $\widehat{v} \in [0, 1]^{d'}$, we have

$$\mathbb{E}\|\widehat{v} - \mu\|_2^2 \geq \frac{d'}{4\,(1 + 2\beta)} - \frac{1}{\beta} \sum_{i=1}^{n} \mathbb{E}\,\langle\widehat{v}, z_i - \mu\rangle. \tag{20}$$

As in [SSTT20], the correlation simplifies as

$$\mathbb{E}\,\langle\widehat{v}, z_i - \mu\rangle = \sum_{j=0}^{d'} \mathbb{E}[\langle\widehat{v}, z_i - \mu\rangle\,|x_i = e_j]\mathbb{P}[x_i = \|\mathcal{X}\|e_j] = \frac{\alpha}{d'}\mathbb{E}[\widehat{v}_j\,(z_i - \mu)_j\,|x_i = \|\mathcal{X}\|e_j]$$

where the last equality follows since $\widehat{v}$ only depends on the coordinate of $z_i$ for which $x_i = 1$. Now, let $\widehat{v}^{\sim i}$ denotes the solution obtained if $z_i$ were replaced by an independent sample. From boundedness, $\left\|\widehat{v}_j\,(z_i - \mu)_j\right\|_\infty \leq 1$. Using differential privacy, we have,

$$\mathbb{E}[\widehat{v}_j\,(z_i - \mu)_j\,|x_i = \|\mathcal{X}\|e_j] \leq (e^\epsilon - 1)\,\mathbb{E}\left[\left|\widehat{v}_j^{\sim i}\,(z_i - \mu)_j\right|\Big|x_i = \|\mathcal{X}\|e_j\right] + \delta \leq (e^\epsilon - 1) + \delta \leq 3\epsilon$$

where the last inequality uses the assumption that $\epsilon \leq 1.2$ and $\delta \leq \epsilon$. Plugging this in Eqn. (20), we get

$$\mathbb{E}\|\widehat{v} - \mu\|_2^2 \geq \frac{d'}{4\,(1 + 2\beta)} - 3\alpha n\epsilon.$$

Plugging the above in Eqn. (19), we get

$$\mathbb{E}[|\langle\widehat{w} - \widetilde{w}, x\rangle|] \geq \frac{B\,\|\mathcal{X}\|\,\alpha}{d'^{(p+1)/p}}\left(\frac{d'}{4\,(1 + 2\beta)} - 3\alpha n\epsilon\right).$$

Using the above, and the bound in Eqn. (17), we get that,

$$\mathbb{E}[L(\widehat{w}; \mathcal{D}) - L(\widetilde{w}; \mathcal{D})] \geq \frac{B\,\|\mathcal{X}\|\,\alpha}{d'^{(p+1)/p}}\left(\frac{d'}{4\,(1 + 2\beta)} - 3\alpha n\epsilon\right) - \frac{2\alpha\beta B\,\|\mathcal{X}\|}{(1 + 2\beta)\,(d')^{1/p}}$$

$$= \frac{B\,\|\mathcal{X}\|\,\alpha}{(1 + 2\beta)\,(d')^{1/p}}\left(\frac{1}{4} - \frac{3\alpha\,(1 + 2\beta)\,n\epsilon}{d'} - 2\beta\right).$$

Now, we set $\beta = \frac{1}{16}$. When rank $\leq 48n\epsilon$ we set $d' = $ rank and $\alpha = \min\left(\frac{d'}{48(1+2\beta)n\epsilon}, 1\right)$. The minimum term becomes $\frac{d'}{48(1+2\beta)n\epsilon}$ and obtain an excess risk lower bound of,

$$\mathbb{E}[L(\widehat{w}; \mathcal{D}) - L(\widetilde{w}; \mathcal{D})] \geq \frac{B\,\|\mathcal{X}\|\,(d')^{(p-1)/p}}{1024n\epsilon}.$$

On the other hand, when rank $> 48n\epsilon$, set $\alpha = 1$ and $d' = \lfloor 48n\epsilon\rfloor$ (so that it is at least 1) to get an excess risk lower bound of,

$$\mathbb{E}[L(\widehat{w}; \mathcal{D}) - L(\widetilde{w}; \mathcal{D})] \geq \min\left(\frac{B\,\|\mathcal{X}\|}{16\,(n\epsilon)^{1/p}}, B\,\|\mathcal{X}\|\right).$$

Finally, note that we can arbitrarily increase the rank of the construction beyond $48n\epsilon$ by adding datapoints with orthogonal feature vectors of small enough magnitude and arbitrary labels. Combining the two lower bounds obtains the claimed bound. $\qquad\square$

**Corollary 4.** *Let $G, B > 0$, $\epsilon \leq 1.2$, $\delta \leq \epsilon$ and $p, q \geq 1$. Let $\mathcal{W} \subset \mathbb{R}^d$ such that for any $w \in \mathcal{W}$, $\|w\|_p \leq B$. For any $(\epsilon, \delta)$-DP algorithm $\mathcal{A}$, there exists a set $\mathcal{Z}$, a distribution $\mathcal{D}$ over $\mathcal{Z}$ and a loss function $w \mapsto \ell(w; z)$, which is convex, G-Lipschitz w.r.t. $\ell_q$ norm for $w \in \mathcal{W}$, for all $z \in \mathcal{Z}$ such that the output of $\mathcal{A}$ on $S \sim \mathcal{D}^n$ (which may not lie in $\mathcal{W}$), satisfies*

$$\mathbb{E}_{\mathcal{A},S}[\mathbb{E}_{z \sim \mathcal{D}} \ell(\mathcal{A}(S); z) - \min_{w \in \mathcal{W}} \mathbb{E}_{z \sim \mathcal{D}} \ell(w; z)] = \Omega\left(GB \min\left(1, \frac{1}{(n\epsilon)^{1/p}}, \frac{d^{(p-1)/p}}{n\epsilon}\right)\right).$$

Using generalization properties of differential privacy, we get the same bound as above for excess empirical risk; see Corollary B.4 in [SSTT20] for details.

## C   Missing Details for Section 5 (Adapting to $\|w^*\|$)

### C.1   Generalized Exponential Mechanism

**Theorem 12.** *[RS15] Let $K > 0$ and $S \in \mathcal{Z}^n$. Let $q_1, ..., q_K$ be functions s.t. for any adjacent datasets $S, S'$ it holds that $|q_j(S) - q_j(S')| \leq \gamma_j : \forall j \in [K]$. There exists an Algorithm, GenExpMech, such that when given sensitivity-score pairs $(\gamma_1, q_1(S)), ..., (\gamma_N, q_N(S))$, privacy parameter $\epsilon > 0$ and confidence parameter $\beta > 0$, outputs $j \in [N]$ such that with probability at least $1 - \beta$ satisfies $q_j(S) \leq \min_{j \in [N]} \left\{q_j(S) + \frac{4\gamma_j \log(N/\beta)}{\epsilon}\right\}$.*

### C.2   Proof of Theorem 7

Note that by assumptions on $\mathcal{A}$, the process of generating $w_1, ..., w_K$ is $(\epsilon/2, \delta/2) - DP$. Furthermore, by Assumption 2 with probability at least $\delta/2$ the sensitivity values passed to GenExpMech bound sensitivity. Thus by the privacy guarantees of GenExpMech and composition we have that the entire algorithm is $(\epsilon, \delta)$-DP.

We now prove accuracy. In order to do so, we first prove that with high probability every $\widetilde{L}_j$ is an upper bound on the true population loss of $w_j$. Specifically, define $\tau_j = \frac{\Delta(B_j) \log(4K/\beta)}{n} + \sqrt{\frac{4\|\mathcal{Y}\|^2 \log(4K/\beta)}{n}}$ (i.e. the term added to each $L(w_j; S_2)$ in Algorithm 4). Note it suffices to prove

$$\mathbb{P}\left[\exists j \in [K] : |L(w_j; S_2) - L(w_j; \mathcal{D})| \geq \tau_j\right] \leq \frac{\beta}{2}. \tag{21}$$

Fix some $j \in [K]$. Note that the non-negativity of the loss implies that $\ell(w_B^*; (x, y)^2) \geq 0$. The excess risk assumption then implies that $\mathbb{E}_{(x,y) \sim \mathcal{D}}\left[\ell(w_j; (x, y))^2\right] \leq 4\|\mathcal{Y}\|^2$, which in turn bounds the variance. Further, with probability at least $1 - \frac{\beta}{4K}$ it holds that for all $(x, y) \in S_2$ that $\ell(w, (x, y)) \leq \Delta_0 + \Delta(B)$. Thus by Bernstein's inequality we have

$$\mathbb{P}\left[|L(w; S_2) - L(w; \mathcal{D})| \geq t\right] \leq \exp\left(-\frac{t^2 n^2}{\Delta(B_j)tn + 4n\|\mathcal{Y}\|^2}\right) + \frac{\beta}{4K}$$

Thus it suffices to set $t = \frac{\Delta(B_j) \log(4K/\beta)}{n} + \sqrt{\frac{4\|\mathcal{Y}\|^2 \log(4K/\beta)}{n}}$ to ensure $\mathbb{P}\left[|L(w; S_2) - L(w; \mathcal{D})| \geq t\right] \leq \frac{\beta}{2K}$. Taking a union bound over all $j \in K$ establishes (21). We now condition on this event for the rest of the proof.

Now consider the case where $j^* \neq 0$ and $\|w^*\| \leq 2^K$. Note that the unconstrained minimizer $w^*$ is the constrained minimizer with respect to any $\mathcal{B}_r$ for $r \geq \|w^*\|$. With this in mind, let $j' = \min_{j \in [K]} \{j : w^* \in \mathcal{B}_{2^j}\}$ (i.e. the index of the smallest ball containing $w^*$). In the following we condition on the event that $\forall j \in [K], j \geq j'$, the parameter vector $w_j$ satisfies excess population risk at most $\text{ERR}(2^j)$. We note by Assumption 2 that this (in addition to the event given in (21)) happens with probability at least $1 - \frac{3\beta}{4}$. By the guarantees of GenExpMech, with probability at least $1 - \beta$

we (additionally) have

$$L(w_{j^*}; \mathcal{D}) \leq L(w_{j^*}; S_2) + \tau_{j^*} \leq \min_{j \in [K]} \left\{ L(w_j; S_2) + \tau_j + \frac{4\Delta(B_j) \log(4K/\beta)}{n\epsilon} \right\}$$

$$\leq L(w_{j'}; S_2) + \tau_{j'} + \frac{4\Delta(B_{j'}) \log(4K/\beta)}{n\epsilon}$$

$$\leq L(w_{j'}; \mathcal{D}) + 2\tau_{j'} + \frac{4\Delta(B_{j'}) \log(4K/\beta)}{n\epsilon}.$$

Since $2^{j'} \leq \max\{2\|w^*\|, 1\}$ we have

$$L(w_{j^*}; \mathcal{D}) - L(w^*; \mathcal{D}) \leq L(w_{j'}; \mathcal{D}) - L(w^*; \mathcal{D}) + 2\tau_{j'} + \frac{4\Delta(B_{j'}) \log(4K/\beta)}{n\epsilon}$$

$$\leq \mathsf{ERR}(2\max\{\|w^*\|, 1\}) + 2\tau_{j'} + \frac{4\Delta(\max\{2\|w^*\|, 1\}) \log(4K/\beta)}{n\epsilon}$$

$$\leq \mathsf{ERR}(2\max\{\|w^*\|, 1\}) + \sqrt{\frac{4\|\mathcal{Y}\|^2 \log(4K/\beta)}{n}} + \frac{5\Delta(\max\{2\|w^*\|, 1\}) \log(4K/\beta)}{n\epsilon}$$

where the second inequality comes from the fact the assumption that $\|w^*\| \leq \|\mathcal{Y}\|^2$. Now note that by the assumption that $\mathsf{ERR}(2^K) \geq \|\mathcal{Y}\|^2$, whenever $\|w^*\| \geq 2^K$ it holds that $\|\mathcal{Y}\|^2 \leq \mathsf{ERR}(\|w^*\|)$. However since the sensitivity-score pair $(0, \|\mathcal{Y}\|^2)$ is passed to $\mathrm{GenExpMech}$, the excess risk of the output is bounded by at most $\|\mathcal{Y}\|^2$ by the guarantees of $\mathrm{GenExpMech}$).

## C.3   Proof of Theorem 8

Let $\widehat{w}$ denote the output of the regularized output perturbation method with boosting and noise and privacy parameters $\epsilon' = \frac{\epsilon}{K}$ and $\delta' = \frac{\delta}{K}$. We have by Theorem 15 that with probability at least $1 - \frac{\beta}{4K}$ that

$$L(\widehat{w}; \mathcal{D}) - L(w^*; \mathcal{D}) = \widetilde{O}\left( \frac{\sqrt{H}B\|\mathcal{X}\|\|\mathcal{Y}\| + \|\mathcal{Y}\|^2}{\sqrt{n}} + \frac{\left( \left(\sqrt{H}B\|\mathcal{X}\|\right)^{4/3}\|\mathcal{Y}\|^{2/3} + \left(\sqrt{H}B\|\mathcal{X}\|\right)^2 \right)}{(n\epsilon)^{2/3}} \right.$$

$$\left. + \frac{\left( \|\mathcal{Y}\|^2 + HB^2\|\mathcal{X}\|^2 \right)}{n\epsilon} + \frac{\left( \|\mathcal{Y}\| + \sqrt{H}B\|\mathcal{X}\| \right)}{\sqrt{n}} \right).$$

Note that this is no smaller than $\|\mathcal{Y}\|^2$ when $B = \Omega\left( \max\left\{ \frac{\|\mathcal{Y}\|\sqrt{n\epsilon}}{\|\mathcal{X}\|\sqrt{H}}, \frac{\|\mathcal{Y}\|^2(n\epsilon)^{2/3}}{\sqrt{H}\|\mathcal{X}\|^2} \right\} \right)$, and thus it suffices to set $K = \Theta\left( \log\left( \max\left\{ \frac{\|\mathcal{Y}\|\sqrt{n}}{\|\mathcal{X}\|\sqrt{H}}, \frac{\|\mathcal{Y}\|^2(n\epsilon)^{2/3}}{\sqrt{H}\|\mathcal{X}\|^2} \right\} \right) \right)$ to satisfy the condition of the Theorem statement.

Let $\sigma_j$ denote the level noise used for when the guess for $\|w^*\|$ is $B_j$. To establish Assumption 2, by Lemma 10 we have that this assumption is satisfies with $\Delta(B) = \|\mathcal{Y}\|^2 + H\|\mathcal{X}\|^2\sigma_j^2 \log(K/\min\{\beta,\delta\})) + HB^2\|\mathcal{X}\|^2$. In particular, we note for the setting of $\sigma_j$ implied by Theorem 15 and the setting of $K$ we have for all $j \in [K]$ that $H\|\mathcal{X}\|^2\sigma_j^2 = \widetilde{O}(\|\mathcal{Y}\|^2)$. Thus $\Delta(B) = \widetilde{O}\left( \|\mathcal{Y}\|^2 + HB^2\|\mathcal{X}\|^2 \right)$. The result then follows from Theorem 7.

## C.4   Stability Results for Assumption 2

**Lemma 9.** *Algorithm 1 run with constraint set $\mathcal{B}_B$ satisfies Assumption 2 with $\Delta(B) = \|\mathcal{Y}\|^2 + HB^2\|\mathcal{X}\|^2$.*

The proof is straightforward using Lemma 3 (provided in the Appendix). For the output perturbation method, we can obtain similar guarantees. Here however, we must account for the fact that the output may not lie in the constraint set. We also remark that the JL-based method (Algorithm 2) can also enjoy this same bound. However, in this case one must apply the norm adaptation method to the intermediate vector $\widetilde{w}$, as $\Phi^\top \widetilde{w}$ may have large norm.

**Lemma 10.** *Algorithm 3 run with parameter $B$ and $\sigma$ satisfies Assumption 2 with $\Delta(B) = \|\mathcal{Y}\|^2 + H\|\mathcal{X}\|^2\sigma^2 \log(K/\delta)) + HB^2\|\mathcal{X}\|^2$*

*Proof.* Note that since $S$ and $S'$ differ in only one point, it suffices to show that for any $(x,y), (x',y')$ that $\ell(\widehat{w}; (x,y)) \leq \|\mathcal{Y}\|^2 + HB^2\|\mathcal{X}\|^2 + H\|\mathcal{X}\|^2\sigma^2 \log(K/\delta)$ and similarly for $\ell(\widehat{w}, (x', y'))$. Let $w \in \mathcal{B}_B$ and let $\widehat{w} = w + b$ where $b \sim \mathcal{N}(0, \mathbb{I}_d\sigma^2)$. We have by previous analysis $\ell(\widehat{w}; (x,y)) \leq \|\mathcal{Y}\|^2 + HB^2\|\mathcal{X}\|^2 + H\langle b, x\rangle^2$. Since $\langle b, x\rangle$ is distributed as a zero mean Gaussian with variance at most $\|\mathcal{X}\|^2\sigma^2$, we have $\mathbb{P}[|\langle b, x\rangle| \geq t] \leq \exp\left(\frac{-t^2}{\|\mathcal{X}\|^2\sigma^2}\right)$. Setting $t = \|\mathcal{X}\|\sigma\log(K/\delta)$ we obtain $\mathbb{P}[|\langle b, x\rangle|^2 \geq \|\mathcal{X}\|^2\sigma^2\log(K/\delta)] \leq \delta/K$. Thus with probability at least $1 - \delta/K$ it holds that $\ell(\widehat{w}; (x,y)) \leq \|\mathcal{Y}\|^2 + HB^2\|\mathcal{X}\|^2 + H\|\mathcal{X}\|^2\sigma^2\log(K/\delta)$. $\square$

## D  Missing Details for Boosting

---
**Algorithm 5** Boosting
---
**Input:** Dataset $S$, loss function $\ell$, Algorithm $\mathcal{A}$, $\widetilde{\sigma}$ privacy parameters $\epsilon, \delta$
 1: Split the dataset $S$ into equally sized chunks $\{S_i\}_{i=1}^{m+1}$
 2: For each $i \in [m+1]$, $\widehat{w}_i = \mathcal{A}\left(S_i, \frac{\epsilon}{2}, \delta\right)$
 3: $i^* = \arg\max_{i \in [m]}\left(-\widehat{L}(\widehat{w}_i; S_{m+1}) + \mathrm{Lap}(0, \widetilde{\sigma})\right)$
**Output:** $\widehat{w}_{i^*}$

---

We state the result of the boosting procedure in a general enough setup so as apply to our proposed algorithms. This leads to additional conditions on the base algorithm since our proposed methods may not produce the output in the constrained set.

**Theorem 13.** *Let $\ell$ be a non-negative, $\widetilde{H}$ smooth, convex loss function. Let $\epsilon, \delta > 0$. Let $\mathcal{A} : (S, \epsilon, \delta) \mapsto \mathcal{A}(S, \epsilon, \delta)$ be an algorithm such that*

  *1. $\mathcal{A}$ satisfies $(\epsilon, \delta)$-DP*

  *2. For any fixed $S$, $\mathcal{A}(S)$ is $\gamma^2$ sub-Gaussian [Ver18]:*

$$\sup_{\|u\|=1} \mathbb{E}\left[\exp\left(\langle \mathcal{A}(S), u\rangle^2/\gamma^2\right)\right] \leq 2$$

  *3. For any fixed $S$, $\mathbb{P}_{(x,y)}[\ell(\mathcal{A}(S); (x,y)) > \Delta(\gamma, \beta)] < \beta$*

  *4. Given a dataset $S$ of $n$ i.i.d. points, $\mathbb{E}[L(\mathcal{A}(S); \mathcal{D}) - \min_{w \in \mathcal{B}_B} L(w; \mathcal{D})] \leq \mathsf{ERR}(n, \epsilon, \gamma)$*

*Let $\widetilde{\sigma}^2 = \frac{4(\|\mathcal{Y}\|^2 + \widetilde{H}\widetilde{\gamma}^2\|\mathcal{X}\|^2)}{n\epsilon}$ and $n_0 = \frac{16\gamma^2 \log^8(4/\beta)\widetilde{H}}{\|\mathcal{Y}\|^2}$. Algorithm 5 with Algorithm $\mathcal{A}$ as input satisfies $(\epsilon, \delta)$-DP. Given a dataset $S$ of $n \geq n_0$ samples, with probability at least $1 - \beta$, the excess risk of its output $\widehat{w}_{i^*}$ is bounded as,*

$$L(\widehat{w}; \mathcal{D}) - L(w^*; \mathcal{D}) \leq \widetilde{O}\Bigg(\mathsf{ERR}\left(\frac{n}{4\log(4/\beta)}, \frac{\epsilon}{2}, \gamma\right) + \frac{2\Delta(\gamma, \beta/2)}{n\epsilon} + \frac{2\Delta\left(\gamma, \frac{\beta}{2n}\right)}{n}$$
$$+ \frac{32\gamma\sqrt{\widetilde{H}}\|\mathcal{Y}\|}{\sqrt{n}} + \frac{16\|\mathcal{Y}\|}{\sqrt{n}} + \frac{128\widetilde{H}\gamma^2}{n}\Bigg).$$

We first establish the following concentration bound for convex $\widetilde{H}$ smooth non-negative functions.

**Lemma 11.** *Let $\ell$ be a convex $\widetilde{H}$ smooth non-negative function. Let $S$ be a dataset of $n$ i.i.d. samples. Let $w$ be a random variable which is $\gamma^2$ sub-Gaussian and independent of $S$ and let $\Delta(\gamma, \beta)$ be such*

*that* $\mathbb{P}_{(x,y)}[\ell(w;(x,y)) > \Delta(\gamma,\beta)] \le \beta$. *Then, with probability at least* $1 - \beta$,

$$\widehat{L}(w;S) \le (1 + T(n,\beta))\, L(w;\mathcal{D}) + U(n,\beta)$$

$$L(w;\mathcal{D}) \le (1 + T(n,\beta))\, \widehat{L}(w;S) + V(n,\beta)$$

*where* $T(n,\beta) := \frac{4\gamma \log(4/\beta)\sqrt{\widetilde{H}}}{\|\mathcal{Y}\|\sqrt{n}}$, $U(n,\beta) := \frac{4\gamma \log(4/\beta)\|\mathcal{Y}\|\sqrt{\widetilde{H}}}{\sqrt{n}} + \frac{\|\mathcal{Y}\|\sqrt{\log(2/\beta)}}{\sqrt{n}}$ *and*
$V(n,\beta) := \frac{4\gamma \log(4/\beta)\sqrt{\widetilde{H}}\|\mathcal{Y}\|}{\sqrt{n}} + \frac{2\Delta\left(\gamma,\frac{\beta}{4n}\right)\log(2/\beta)}{n} + \frac{\|\mathcal{Y}\|\sqrt{\log(2/\beta)}}{\sqrt{n}} + \frac{48\widetilde{H}\gamma^2 \log^2(4/\beta)}{n}$.

*Proof.* With probability at least $1 - \frac{\beta}{4}$, for each $(x,y) \in S, \ell(w;(x,y)) \le \Delta\left(\gamma,\frac{\beta}{4n}\right)$. We condition on this event and apply Bernstein inequality to the random variable $L(w;\mathcal{D}) - \widehat{L}(w;S)$:

$$\mathbb{P}\left[\left|L(w;\mathcal{D}) - \widehat{L}(w;S)\right| > t\right] \le \exp\left(-\frac{3nt^2}{6n\mathbb{E}[\left(L(w;\mathcal{D}) - \widehat{L}(w;S)\right)^2] + 2\Delta\left(\gamma,\frac{\beta}{4n}\right)t}\right)$$

This gives us that

$$\left|L(w;\mathcal{D}) - \widehat{L}(w;S)\right| \le \frac{\Delta\left(\gamma,\frac{\beta}{4n}\right)\log(2/\beta)}{n} + \sqrt{\mathbb{E}\left(L(w;\mathcal{D}) - \widehat{L}(w;S)\right)^2 \log(2/\beta)} \quad (22)$$

The term $\mathbb{E}[\left(L(w;\mathcal{D}) - \widehat{L}(w;S)\right)^2 = \frac{1}{n}\mathbb{E}[(\ell(w;(x,y)) - \mathbb{E}[\ell(w;(x,y))])^2] \le \frac{1}{n}\mathbb{E}[(\ell(w;(x,y)))^2]$. Now,

$$\mathbb{E}[(\ell(w;(x,y)))^2] \le 2\mathbb{E}[(\ell(w;(x,y)) - \ell(0;(x,y))^2] + 2\mathbb{E}[(\ell(0;(x,y)))^2]$$

$$\le 2\mathbb{E}[(\langle\nabla\ell(w;(x,y)), w\rangle)^2] + 2\|\mathcal{Y}\|^2$$

where the last step follows from convexity. We now use the fact that $w$ is $\gamma^2$-sub-Gaussian, therefore $\langle\nabla\ell(w;(x,y)), w\rangle \le \gamma\sqrt{\log(4/\beta)}\|\nabla\ell(w;(x,y))\|$ with probability at least $1 - \beta/4$. We now use self-bounding property of non-negative smooth functions to get,

$$\mathbb{E}[(\ell(w;(x,y)))^2] \le 2\mathbb{E}[\|\nabla\ell(w;(x,y))\|^2 \gamma^2 \log(4/\beta) + 2\|\mathcal{Y}\|^2$$

$$\le 8\widetilde{H}\mathbb{E}[\ell(w;(x,y))]\gamma^2 \log(4/\beta) + 2\|\mathcal{Y}\|^2$$

$$= 8\widetilde{H}L(w;\mathcal{D})\gamma^2 \log(4/\beta) + 2\|\mathcal{Y}\|^2$$

Plugging the above in Eqn (22) gives us,

$$\left|L(w;\mathcal{D}) - \widehat{L}(w;S)\right| \le \frac{\Delta\left(\gamma,\frac{\beta}{4n}\right)\log(2/\beta)}{n} + 4\sqrt{\frac{\left(\widetilde{H}L(w;\mathcal{D})\gamma^2 \log(4/\beta) + \|\mathcal{Y}\|^2\right)\log(1/\beta)}{n}}$$

$$\le \frac{\Delta\left(\gamma,\frac{\beta}{4n}\right)\log(2/\beta)}{n} + 4\sqrt{\frac{\widetilde{H}L(w;\mathcal{D})}{n}}\gamma\log(4/\beta) + \frac{\|\mathcal{Y}\|\sqrt{\log(2/\beta)}}{\sqrt{n}}.$$

$$(23)$$

A simple application of AM-GM inequality gives,

$$\widehat{L}(w;S) \le \left(1 + \frac{4\gamma\log(4/\beta)\sqrt{\widetilde{H}}}{\|\mathcal{Y}\|\sqrt{n}}\right)L(w;\mathcal{D}) + \frac{4\gamma\log(4/\beta)\|\mathcal{Y}\|\sqrt{\widetilde{H}}}{\sqrt{n}} + \frac{\|\mathcal{Y}\|\sqrt{\log(2/\beta)}}{\sqrt{n}}$$

This proves the first part of the lemma. For the second part, we use the following fact about non-negative numbers $A, B, C$ [BBL03] (see after proof of Theorem 7)

$$A \leq B + C\sqrt{A} \implies A \leq B + C^2 + \sqrt{B}C$$

Thus, from Eqn. (23),

$$L(w; \mathcal{D}) \leq \widehat{L}(w; S) + \frac{\Delta\left(\gamma, \frac{\beta}{4n}\right) \log\left(2/\beta\right)}{n} + \frac{\|\mathcal{Y}\|\sqrt{\log\left(2/\beta\right)}}{\sqrt{n}} + \frac{16\widetilde{H}\gamma^2 \log^2\left(4/\beta\right)}{n}$$

$$+ \frac{4\gamma \log\left(4/\beta\right)\sqrt{\widetilde{H}}}{\sqrt{n}} \left( \sqrt{\widehat{L}(w; S)} + \sqrt{\frac{\Delta\left(\gamma, \frac{\beta}{4n}\right) \log\left(2/\beta\right)}{n}} + \sqrt{\frac{\|\mathcal{Y}\|\sqrt{\log\left(2/\beta\right)}}{\sqrt{n}}} \right)$$

$$\leq \widehat{L}(w; S) + \frac{4\gamma \log\left(4/\beta\right)\sqrt{\widetilde{H}\widehat{L}(w; S)}}{\sqrt{n}} + \frac{\Delta\left(\gamma, \frac{\beta}{4n}\right) \log\left(2/\beta\right)}{n} + \frac{\|\mathcal{Y}\|\sqrt{\log\left(2/\beta\right)}}{\sqrt{n}}$$

$$+ \frac{16\widetilde{H}\gamma^2 \log^2\left(4/\beta\right)}{n} + \frac{4\gamma\sqrt{\widetilde{H}\Delta\left(\gamma, \frac{\beta}{4n}\right)} \log^{3/2}\left(4/\beta\right)}{n} + \frac{4\gamma\sqrt{\widetilde{H}\|\mathcal{Y}\|} \log^{5/4}\left(2/\beta\right)}{n^{3/4}}$$

$$\leq \left( 1 + \frac{4\gamma \log\left(4/\beta\right)\sqrt{\widetilde{H}}}{\|\mathcal{Y}\|\sqrt{n}} \right) \widehat{L}(w; S) + \frac{4\gamma \log\left(4/\beta\right)\|\mathcal{Y}\|\sqrt{\widetilde{H}}}{\sqrt{n}} + \frac{\Delta\left(\gamma, \frac{\beta}{4n}\right) \log\left(2/\beta\right)}{n}$$

$$+ \frac{\|\mathcal{Y}\|\sqrt{\log\left(2/\beta\right)}}{\sqrt{n}} + \frac{16\widetilde{H}\gamma^2 \log^2\left(4/\beta\right)}{n} + \frac{4\gamma\sqrt{\widetilde{H}\Delta\left(\gamma, \frac{\beta}{4n}\right)} \log^{3/2}\left(4/\beta\right)}{n}$$

$$+ \frac{4\gamma\sqrt{\widetilde{H}\|\mathcal{Y}\|} \log^{5/4}\left(2/\beta\right)}{n^{3/4}}$$

where the last inequality follows from AM-GM inequality. Simplifying the expressions yields the claimed bound. $\qquad \square$

*Proof of Theorem 13.* Since the models $\{\widehat{w}_i\}_{i=1}^m$ are trained on disjoint datasets, by parallel composition $\{\widehat{w}_i\}_{i=1}^m$ satisfies $\left(\frac{\epsilon}{2}, \frac{\delta}{2}\right)$-DP. We know that probability at least $1 - \frac{\delta}{2}$, $\ell(w; (x, y)) \leq \Delta\left(\gamma, \frac{\delta}{2}\right)$. Thus conditioning on this event, from the guarantee of the report noisy max procedure, we have that it satisfies $\left(\frac{\epsilon}{2}\right)$-DP. The privacy proof thus follows from absorbing the failure probability into $\delta$ part and adaptive composition.

We proceed to the utility part. Let $\widetilde{w}$ be the model among $\{\widehat{w}_i\}_{i=1}^m$ with minimum empirical error on the set $S_{m+1}$. The excess risk of each $\widehat{w}_i$ is bounded as,

$$\mathbb{E}[L(\widehat{w}_i; \mathcal{D})] - L(w^*; \mathcal{D}) \leq \mathsf{ERR}\left(\frac{n}{m+1}, \frac{\epsilon}{2}, \gamma\right)$$

From Markov's inequality, with probability at least $3/4$, $L(\widehat{w}_i; \mathcal{D}) \leq L(w^*; \mathcal{D}) + 4\mathsf{ERR}\left(\frac{n}{m+1}, \frac{\epsilon}{2}\right)$. From independence of $\{w_i\}_{i=1}^m$, with probability at least $1 - 1/4^m = 1 - \frac{\beta}{4}$, there exists one model, say $\widehat{w}_{i^*}$, such $L(\widehat{w}_{i^*}; \mathcal{D}) \leq L(w^*; \mathcal{D}) + 4\mathsf{ERR}\left(\frac{n}{m+1}, \frac{\epsilon}{2}\right)$.

Also, from the guarantee of Report-Noisy-Max, we have that with probability at least $1 - \beta/4$

$$L(\widehat{w}; S_{m+1}) \leq L(\widetilde{w}; S_{m+1}) + \frac{\Delta(\gamma, \beta/4)(m+1)\log^2\left(4m/\delta\right)}{n\epsilon}$$

Now, we apply Lemma 11. From a union bound, with probability at least $1 - \frac{\beta}{2}$, all $\{w_i\}_{i=1}^m$ satisfy the inequalities in Lemma 11 with $\beta$ substituted as $\frac{\beta}{2m}$.

Thus, for the output $\widehat{w}$, probability at least $1 - \frac{\beta}{2}$,

$L(\widehat{w}; \mathcal{D})$

$$\leq \left(1 + T\left(\frac{n}{m+1}, \frac{\beta}{2m}\right)\right) L(\widehat{w}; S_{m+1}) + V\left(\frac{n}{(m+1)}, \frac{\beta}{2m}\right)$$

$$\leq \left(1 + T\left(\frac{n}{m+1}, \frac{\beta}{2m}\right)\right) L(\widetilde{w}; S_{m+1}) + \left(1 + T\left(\frac{n}{m+1}, \frac{\beta}{2m}\right)\right) \frac{\Delta(\gamma, \beta/4)(m+1)\log^2(4m/\delta)}{n\epsilon}$$

$$+ V\left(\frac{n}{(m+1)}, \frac{\beta}{2m}\right)$$

$$\leq \left(1 + T\left(\frac{n}{m+1}, \frac{\beta}{2m}\right)\right) L(w_{i^*}; S_{m+1}) + \frac{2\Delta(\gamma, \beta/4)(m+1)\log^2(4m/\delta)}{n\epsilon} + V\left(\frac{n}{(m+1)}, \frac{\beta}{2m}\right)$$

where in the above we use that $T\left(\frac{n}{m+1}\right) \leq 1$ given the lower bound on $n$ and the setting of $m$. Furthermore, the last inequality follows because $\widetilde{w}$ has lowest empirical risk on $S_{m+1}$. Let $W(n, m, \beta) = \frac{2\Delta(\gamma, \beta/4)(m+1)\log^2(4m/\delta)}{n\epsilon} + V\left(\frac{n}{(m+1)}, \frac{\beta}{2m}\right)$. We now apply the other guarantee in Lemma 11 and the fact that $w_{i^*}$ has small excess risk. With probability at least $1 - \delta/2$,

$L(\widehat{w}; \mathcal{D})$

$$\leq \left(1 + T\left(\frac{n}{m+1}, \frac{\beta}{2m}\right)\right)^2 L(w_{i^*}; \mathcal{D}) + \left(1 + T\left(\frac{n}{m+1}, \frac{\beta}{2m}\right)\right) U\left(\frac{n}{m+1}, \frac{\beta}{2m}\right) + W(n, m, \beta)$$

$$\leq L(w^*; \mathcal{D}) + 2T\left(\frac{n}{m+1}, \frac{\beta}{2m}\right) L(w^*; \mathcal{D}) + 4\mathrm{ERR}\left(\frac{n}{m+1}, \frac{\epsilon}{2}\right)$$

$$+ 8T\left(\frac{n}{m+1}, \frac{\beta}{2m}\right) 4\mathrm{ERR}\left(\frac{n}{m+1}, \frac{\epsilon}{2}\right) + 2U\left(\frac{n}{m+1}, \frac{\beta}{2m}\right) + W(n, m, \beta)$$

Let $X(n, m, \beta) = 4\mathrm{ERR}\left(\frac{n}{m+1}, \frac{\epsilon}{2}\right) + 8T\left(\frac{n}{m+1}, \frac{\beta}{2m}\right) 4\mathrm{ERR}\left(\frac{n}{m+1}, \frac{\epsilon}{2}\right) + 2U\left(\frac{n}{m+1}, \frac{\beta}{2m}\right) + W(n, m, \beta)$. Note that $m = 4\log(4/\beta)$ and $T\left(\frac{n}{m+1}, \frac{\beta}{2m}\right) \leq \frac{16\gamma\log^4(2/\beta)\sqrt{H}}{\|\mathcal{Y}\|\sqrt{n}}$. Substituting this and using the fact that, $L(w^*; \mathcal{D}) \leq L(0; \mathcal{D}) \leq \|\mathcal{Y}\|^2$, we get that with probability at least $1 - \delta$,

$$L(\widehat{w}; \mathcal{D}) \leq L(w^*; \mathcal{D}) + \frac{16\gamma\log^4(2/\beta)\sqrt{H}\|\mathcal{Y}\|}{\sqrt{n}} + X(n, 4\log(2/\beta), \beta)$$

Substituting and simplifying the $X(n, 4\log(4/\beta), \beta)$ we have that

$X(n, 4\log(2/\beta), \beta)$

$$\leq 12\mathrm{ERR}\left(\frac{n}{4\log(4/\beta)}, \frac{\epsilon}{2}, \gamma\right) + \frac{2\Delta(\gamma, \beta/4)\log^3(4/\beta)\log(2/\delta)}{n\epsilon} + \frac{2\Delta\left(\gamma, \frac{\beta}{2n}\right)\log^4(8/\beta)}{n}$$

$$+ \frac{16\gamma\log^4(8/\beta)\sqrt{\widetilde{H}}\|\mathcal{Y}\|}{\sqrt{n}} + \frac{16\|\mathcal{Y}\|\log^4(8/\beta)}{\sqrt{n}} + \frac{128\widetilde{H}\gamma^2\log^4(8/\beta)}{n}$$

Hence, with probability at least $1 - \beta$,

$L(\widehat{w}; \mathcal{D})$

$$\leq L(w^*; \mathcal{D}) + 12\mathrm{ERR}\left(\frac{n}{4\log(4/\beta)}, \frac{\epsilon}{2}, \gamma\right) + \frac{2\Delta(\gamma, \beta/4)\log^3(4/\beta)\log(2/\delta)}{n\epsilon} + \frac{2\Delta\left(\gamma, \frac{\beta}{2n}\right)\log^4(8/\beta)}{n}$$

$$+ \frac{32\gamma\log^4(8/\beta)\sqrt{\widetilde{H}}\|\mathcal{Y}\|}{\sqrt{n}} + \frac{16\|\mathcal{Y}\|\log^4(8/\beta)}{\sqrt{n}} + \frac{128\widetilde{H}\gamma^2\log^4(8/\beta)}{n}$$

$\square$

## D.1 Boosting the JL Method

**Theorem 14.** *The boosting procedure (Algorithm 5) using the JL method (Algorithm 2) as Algorithm $\mathcal{A}$ satisfies $(\epsilon, \delta)$-DP, and with probability at least $1 - \beta$, its output $\widehat{w}_{i^*}$ has excess risk,*

$$
L(\widehat{w}_{i^*}; \mathcal{D}) - L(w^*; \mathcal{D}) \leq \widetilde{O}\left( \frac{\sqrt{H}B \, \|\mathcal{X}\| \, \|\mathcal{Y}\|}{\sqrt{n}} + \frac{\left( \left( \sqrt{H}B \, \|\mathcal{X}\| \right)^{4/3} \|\mathcal{Y}\|^{2/3} + \left( \sqrt{H}B \, \|\mathcal{X}\| \right)^2 \right)}{(n\epsilon)^{2/3}} \right.
$$

$$
\left. + \frac{\left( \|\mathcal{Y}\|^2 + HB^2 \|\mathcal{X}\|^2 \right)}{n\epsilon} + \frac{\left( \|\mathcal{Y}\| + \sqrt{H}B \|\mathcal{X}\| \right)}{\sqrt{n}} \right)
$$

*Proof.* For the JL method, we consider the boosting procedure in the $k$ dimension space – this is only for the sake of analysis and the algorithm remains the same. In particular, consider the distribution to $\Phi\mathcal{D}$ which, when sampled from, gives the data point $(\Phi x, y)$ where $(x, y) \sim \mathcal{D}$.

Suppose the boosting procedure gives the following $k$ dimensional models: $\widetilde{w}_1, \cdots \widetilde{w}_m$; note that the norm of all these are bounded by $2B$. Let $\widetilde{w}^* \in \arg\min_{\|w\| \leq 2B} L(w; \Phi\mathcal{D})$. Since the outputs satisfy $\|\widetilde{w}_i\| \leq 2B$, the sub-Gaussian parameter $\gamma = O(B)$. We now compute the other parameter $\Delta(\gamma, \beta)$, which is the high probability bound on loss. Note that for a fixed data point $(\Phi x, y)$, an $H$ smooth, non-negative, bounded at zero loss, at any point $w$ s.t. $\|w\| \leq 2B$ is upper bounded by $2\left( \|\mathcal{Y}\|^2 + 4B^2 H \|\Phi x\|^2 \right)$. From the JL guarantee, with the given $k$, the term $\|\Phi x\|^2 \leq 2 \|\mathcal{X}\|$ with probability at least $1 - \beta/4$. This gives us $\Delta(2B, \beta) = 2\left( \|\mathcal{Y}\|^2 + 16 \|\mathcal{X}\|^2 B^2 \right)$.

We now invoke 13 substituting $\Delta$, $\gamma$ and $\widetilde{H} = H\|\mathcal{X}\|^2$ to get that with probability at least $1 - \frac{\beta}{2}$ output satisfies,

$$
L(\widetilde{w}_{i^*}; \Phi\mathcal{D}) \leq L(\widetilde{w}^*; \Phi\mathcal{D}) + \frac{128B\sqrt{H}\|\mathcal{X}\|\|\mathcal{Y}\| \log^4 (8/\beta)}{\sqrt{n}} + \alpha\left( \frac{n}{4 \log((8/\beta))}, \frac{\epsilon}{2}, 2B \right)
$$

$$
+ \frac{128 \left( \|\mathcal{Y}\|^2 + HB^2\|\mathcal{X}\|^2 \right) \log^4 (8/\beta)}{n\epsilon} + \frac{128 \left( \|\mathcal{Y}\| + \sqrt{H}B\|\mathcal{X}\| \right) \log^4 (8/\beta)}{\sqrt{n}}
$$

Define $W := \frac{128B\sqrt{H}\|\mathcal{X}\|\|\mathcal{Y}\| \log^4 (8/\beta)}{\sqrt{n}} + \frac{128\left( \|\mathcal{Y}\|^2 + HB^2\|\mathcal{X}\|^2 \right) \log^4 (8/\beta)}{n\epsilon} + \frac{128\left( \|\mathcal{Y}\| + \sqrt{H}B\|\mathcal{X}\| \right) \log^4 (8/\beta)}{\sqrt{n}}$. The excess risk of the final output $\widehat{w}_{i^*} = \Phi^\top \widetilde{w}_{i^*}$ is bounded as,

$$
L(\widehat{w}_{i^*}; \mathcal{D}) - L(w^*; \mathcal{D}) = L(\widetilde{w}_{i^*}; \Phi\mathcal{D}) - L(w^*; \mathcal{D})
$$

$$
\leq L(\widetilde{w}^*; \Phi\mathcal{D}) + \alpha\left( \frac{n}{4 \log (8/\beta)}, \frac{\epsilon}{2}, 2B \right) + W - L(w^*; \mathcal{D})
$$

$$
\leq L(\Phi w^*; \Phi\mathcal{D}) + \alpha\left( \frac{n}{4 \log (8/\beta)}, \frac{\epsilon}{2}, 2B \right) + W - L(w^*; \mathcal{D})
$$

$$
\leq \alpha\left( \frac{n}{4 \log (8/\beta)}, \frac{\epsilon}{2}, 2B \right) + W + \frac{H}{2} |\langle \Phi x, \Phi w^* \rangle - \langle x, w^* \rangle|^2
$$

where the last inequality follows from smoothness and that $\nabla L(w^*; \mathcal{D}) = 0$. Finally, from the JL property, with probability at least $1 - \frac{\beta}{2}$, $|\langle \Phi x, \Phi w^* \rangle - \langle x, w^* \rangle|^2 \leq \alpha^2 \|w^*\|^2 \|\mathcal{X}\|^2$. Combining and substituting the values of $\alpha$ and $\alpha\left( \frac{n}{4 \log(8/\beta)}, \frac{\epsilon}{2} \right)$ from Theorem 2 gives the claimed result. $\square$

## D.2 Boosting Output Perturbation Method

**Theorem 15.** *The boosting procedure (Algorithm 5) using the output perturbation method (Algorithm 3) as input Algorithm $\mathcal{A}$ satisfies $(\epsilon, \delta)$-DP, and with probability at least $1 - \beta$, its output $\widehat{w}_{i^*}$ has*

*excess risk,*

$$L(\widehat{w}_{i^*}; \mathcal{D}) - L(w^*; \mathcal{D}) \le \widetilde{O}\left( \frac{\sqrt{H}B \, \|\mathcal{X}\| \, \|\mathcal{Y}\| + \|\mathcal{Y}\|^2}{\sqrt{n}} + \frac{\left( \left( \sqrt{H}B \, \|\mathcal{X}\| \right)^{4/3} \|\mathcal{Y}\|^{2/3} + \left( \sqrt{H}B \, \|\mathcal{X}\| \right)^2 \right)}{(n\epsilon)^{2/3}} \right.$$

$$\left. + \frac{\left( \|\mathcal{Y}\|^2 + HB^2 \|\mathcal{X}\|^2 \right)}{n\epsilon} + \frac{\left( \|\mathcal{Y}\| + \sqrt{H}B \|\mathcal{X}\| \right)}{\sqrt{n}} \right)$$

*Proof.* Firstly, note that $\widetilde{H} = H\|\mathcal{X}\|^2$. We now compute $\gamma$ and $\Delta$ to invoke Theorem 13. Since the algorithm simply adds Gaussian noise of variance $\sigma^2 \mathbb{I}_d$ to a vector $\widetilde{w}$ where $\|\widetilde{w}\| \le B$, we have that $\gamma^2 \le B^2 + \sigma^2$. For the bound on loss parameter $\Delta$, by direct computation, $\Delta(\gamma, \beta) \le 2\left( \|\mathcal{Y}\|^2 + H \langle \widetilde{w} + \xi, x \rangle^2 \right) \le 2\|\mathcal{Y}\|^2 + 2HB^2\|\mathcal{X}\|^2 + 2H \langle \xi, x \rangle^2 \le 2\|\mathcal{Y}\|^2 + 2HB^2\|\mathcal{X}\|^2 + 2H\sigma^2 \log(1/\beta)$ where the last inequality follows since $\langle \xi, x \rangle \sim \mathcal{N}(0, \|x\|^2)$. Plugging these and the value of $\sigma^2$ from Theorem 3 into Theorem 13 gives the claimed bound. $\square$

# E  Non-private Lower Bounds

We first note a simple one-dimensional lower bound.

**Theorem 16.** *(Implicit in [SST10]) Let $\|\mathcal{X}\| \ge 1$ and $H \ge 2$. For any Algorithm $\mathcal{A}$, there exists a 1-dimensional $H$-smooth non-negative GLM, $\ell : \mathbb{R} \times (\mathcal{X} \times \mathcal{Y}) \mapsto \mathbb{R}$, and a distribution $\mathcal{D}$ over $(\mathcal{X} \times \mathcal{Y})$ with $\|w^*\| = \Theta(\min\{\|\mathcal{Y}\|, B\})$ such that the excess population risk of the output of $\mathcal{A}$ when run on $S \sim \mathcal{D}^n$ is lower bounded as $\Omega\left( \frac{\min\{\|\mathcal{Y}\|, B\|\mathcal{X}\|\}}{\sqrt{n}} \right)$.*

We remark that this requires a slight modification of the example used in [SST10]. Specifically, therein the loss function is defined as

$$\ell(w, (x, y)) = \begin{cases} (w - y)^2 & |w - y| \le \frac{1}{2} \\ |w - y| - 1/4 & |w - y| \ge \frac{1}{2} \end{cases}$$

with $y \in \{\pm 1\}$ and $x = 1$. Our statement is obtained by setting the domain of labels as $\{\pm \min\{B, \|\mathcal{Y}\|\}\}$. A reduction in [Sha15] enables lower bounds from problem instances with general $\|\mathcal{X}\|$ and $\|w^*\| = R$ to instances with $\|\mathcal{X}\| = 1$ to $\|w^*\| = R\|\mathcal{X}\|$.

We now show that the lower bounds presented in [Sha15] to the unconstrained setting. We start by stating the original bound from [Sha15] which holds for squared loss.

**Theorem 17.** *Let $\ell(w, (x, y)) = \frac{1}{2}(\langle w, x \rangle - y)^2$ be the squared loss function and $B > 0$. Then for any algorithm, $\mathcal{A}$, there exists a distribution $\mathcal{D}$ over $(\mathcal{X} \times \mathcal{Y})$ and a constant $C$ such that for $S \sim \mathcal{D}^n$ it holds that $\mathbb{E}\left[ L(\mathcal{A}(S); \mathcal{D}) - L(w_B^*; \mathcal{D}) \right] \ge C \min\left\{ \|\mathcal{Y}\|^2, \frac{B^2\|\mathcal{X}\|^2 + d\|\mathcal{Y}\|^2}{n}, \frac{B\|\mathcal{Y}\|\|\mathcal{X}\|}{\sqrt{n}} \right\}$, where $w_B^* = \arg\min_{w \in \mathcal{B}_B} \{L(w; \mathcal{D})\}$.*

We now make three observations. First we note that this Theorem holds even for $\mathcal{A}(S) \notin \mathcal{B}_B$. Second we note that the construction of the distribution in the above Theorem is such that $\min_{w^* \in \mathcal{B}_B} \{L(w; \mathcal{D})\} = \min_{w^* \in \mathbb{R}^d} \{L(w; \mathcal{D})\}$. Finally, note that $\frac{H}{2}(\langle w, x \rangle - \frac{y}{\sqrt{H}})^2 = \frac{1}{2}(\sqrt{H} \langle w, x \rangle - y)^2$. This gives the following corollary.

**Corollary 5.** *Let $B > 0$. Then for any algorithm, $\mathcal{A}$, there exists a distribution $\mathcal{D}$ over $(\mathcal{X} \times \mathcal{Y})$ and an $H$-smooth non-negative GLM, $\ell : \mathbb{R}^d \times (\mathcal{X} \times \mathcal{Y}) \mapsto \mathbb{R}$, with minimizer $w^* = \arg\min_{w \in \mathbb{R}^d} \{L(w; \mathcal{D})\}$ such that $\|w^*\| = B$ and for $S \sim \mathcal{D}^n$ it holds that*

$$\mathbb{E}\left[ L(\mathcal{A}(S); \mathcal{D}) - L(w^*; \mathcal{D}) \right] = \Omega\left\{ \min\left\{ \|\mathcal{Y}\|^2, \frac{HB^2\|\mathcal{X}\|^2 + d\|\mathcal{Y}\|^2}{n}, \frac{\sqrt{H}B\|\mathcal{Y}\|\|\mathcal{X}\|}{\sqrt{n}} \right\} \right\}.$$

# F    Additional Results

**Lemma 12.** *Let $\Phi \in \mathbb{R}^{d \times k}$ be a random matrix such that for any $u \in \mathbb{R}^d$, with probability at least $1 - \beta$, $(1 - \alpha) \|u\|^2 \leq \|\Phi u\|^2 \leq (1 + \alpha) \|u\|^2$. Then for any $u, v$, with probability at least $1 - 2\beta$, $|\langle \Phi u, \Phi v \rangle - \langle u, v \rangle| \leq \alpha \|u\| \|v\|$.*

*Proof.* Firstly, note that it suffices to prove the result for unit vectors $u$ and $v$. From the norm preservation result, with probability at least $1 - 2\beta$, we have that,

$$(1 - \alpha) \|u + v\|^2 \leq \|\Phi(u + v)\| \leq (1 + \alpha) \|u + v\|^2$$
$$(1 - \alpha) \|u - v\|^2 \leq \|\Phi(u - v)\| \leq (1 + \alpha) \|u - v\|^2$$

Therefore, we have

$$
\begin{aligned}
\langle \Phi u, \Phi v \rangle &= \frac{1}{4} \left( \|\Phi(u + v)\|^2 - \|\Phi(u - v)\|^2 \right) \\
&\leq \frac{1}{4} \left( (1 + \alpha) \|u + v\|^2 - (1 - \alpha) \|u - v\|^2 \right) \\
&\leq \langle u, v \rangle + \alpha
\end{aligned}
$$

This gives us that $\langle \Phi u, \Phi v \rangle \leq \langle u, v \rangle + \alpha$. The other inequality follows in the same way. $\square$