# OpenReview forum: "Differentially Private Generalized Linear Models Revisited"
_NeurIPS.cc/2022/Conference — NeurIPS 2022 Accept_

### Official Review · Reviewer_pgk1 · 2022-07-11

**Rating:** 7
**Confidence:** 5
**Soundness:** 4 excellent
**Presentation:** 3 good
**Contribution:** 3 good

**Summary:**

The paper studies the problem of differentially private generalized linear models (GLM) with convex losses. The main contribution of the paper are new upper bounds on the rate for smooth and non-negative functions that improve over existing bounds. The authors also prove lower bounds which show that these bounds are nearly-optimal up to factors that depend on the norm of the optimal solution. The authors also consider the setting of Lipschitz functions and show that their algorithms achieve the optimal rate up to log factors.

**Questions:**

See weaknesses.

**Limitations:**

Yes

**Strengths And Weaknesses:**

Strengths
The paper studies an interesting and important problem and improves over the previously known rates for this problem under various settings.

Weaknesses
1.	The rates for the smooth case depend on the dimension d (not the rank as in the Lipschitz case). While the authors show tight lower bounds up to factors that depend on |w*|, this lower bound is probably obtained by taking rank=d and therefore the upper bound may not have tight dependence on the rank. It is important to clarify this in the paper.
2.	One minor weakness is that all the algorithms in the paper are somewhat standard as similar algorithms have been used in the literature. However, the obtained results and some of the techniques are interesting (such as using average stability instead of uniform stability to improve the rates), therefore I don’t consider this to be a real limitation.
3.	The rates obtained for the smooth case are very similar to the rates for DP-SCO in ell_1 geometry [AFKT21, BGN21] which also have a phase transition. I’m not sure if there is any fundamental connection here but it may be useful to explore this a little and comment on this.
4.	Why didn’t the authors use the algorithms from “private selection from private candidates” [LT18] for adaptivity to |w*|? This will only require to privatize the score functions (as [LT18] assumes a non-private one) but may be simpler than redoing everything from scratch.
More minor comments:
1.	Text in table 1 is too small and hard to read
2.	Algorithm 1: gradient symbol is missing in line 4

References:
[AFKT21] Private Stochastic Convex Optimization: Optimal Rates in ℓ1 Geometry
[BGN21] Non-Euclidean Differentially Private Stochastic Convex Optimization
[LT18] Private selection from private candidates

---

> ### Author Response · Authors · 2022-08-02
> **Review Response**
>
> Thank you for these observations. We will add a clarification about the rank bound in the paper. We also observed this coincidence with [AFKT21], but could not find any principled explanation for this coincidence, and so did not comment on it in the paper. With regards to [LT18], we in fact use the generalized exponential mechanism as a black box result, which is similar to the method you cite from [LT18]. We will add [LT18] as an additional citation.
>
> Thank you also for pointing out these formatting issues. We will fix the formatting issue with Table 1 and the typo you mentioned.

---

### Official Review · Reviewer_VnwD · 2022-07-11

**Rating:** 6
**Confidence:** 3
**Soundness:** 3 good
**Presentation:** 4 excellent
**Contribution:** 4 excellent

**Summary:**

This paper revisits DP GLM and it provides a thorough theoretical analysis about different cases. Both upper and lower bounds are provided.

Based on their DP definition, it is a global DP model with $(\epsilon, \delta)$ as privacy parameters i.e., approximated DP . All the key results seem to have high probability guarantees. I am wondering if it is possible to extend the result to pure-DP, i.e., the $(\epsilon, 0)$ one?


**Questions:**

See Summary

**Ethics Review Area:**

["I don’t know"]

**Limitations:**

See Summary

**Strengths And Weaknesses:**

Strength: revisit the DP GLMs; thorough analysis are provided.
Weakness: n/a

---

> ### Author Response · Authors · 2022-08-02
> **Review Response**
>
> Thanks for your review. Pure DP analogs are likely possible using different techniques, but this often leads to worse rates. For this reason we did not consider pure DP algorithms in our work.

---

### Official Review · Reviewer_qYvP · 2022-07-11

**Rating:** 7
**Confidence:** 2
**Soundness:** 3 good
**Presentation:** 3 good
**Contribution:** 3 good

**Summary:**

This paper studies the sample complexity of learning differentially private generalized linear models and proposes upper and lower bounds for this problem. For smooth non-negative GLMs, their rates exhibit an interesting learning phenomenon in terms of the privacy budget and dimension. For high privacy budget (low-dimension), a novel analysis of noisy GD is shown to achieve the upper bound. For  low-privacy regime, the upper bound is achieved by an algorithm that uses random projection to derive low-dimensional projections of the data. When the GLM loss function is Lipschitz, they close an open gap in literature in terms of lower and upper bound.

**Questions:**

--
Are projection steps necessary for proof? Alternatively, did the authors consider a Frank-Wolfe version of Algorithm 1/2? If so, are the guarantees very weak?

--
Have the authors tried to consider an adaptive grid search algorithm for Algorithm~4?

**Limitations:**

--
Computational implementation seems limited, thereby limiting applicability.

**Strengths And Weaknesses:**

--
Strength:
The paper proposes rate-optimal differentially-private algorithms for the learning GLM loss functions. To my knowledge, this is much harder than regression because of the curvature of the loss function. The results presented use low-magnitude noise and applicable to a wide variety of loss-function curvature that occur in practice. From a purely technical perspective, some of the intermediate results are of independent interest. Results for several applications of interest are also presented.

--
Weakness:
A major limitation of Algorithm 1 and 2 is its reliance on projection-based techniques for restricting the iterate to the constraint set. As is well-established this is very difficult to implement in practice and is computationally burdensome. Similarly, Algorithm 4 is a static gird search, which could be made adaptive to the dataset.

 Very strong theoretical results, limited computational applicability.

---

> ### Author Response · Authors · 2022-08-02
> **Review Response**
>
> Thank you for your feedback. With regards to the projection sets and Frank-Wolfe, we note that the constraint set is simply an $\ell_2$ ball around the origin of a specified radius, so projection can be done very efficiently by simply scaling the vector. About the question on using a data-adaptive grid search, note that the overhead of our static grid search, statistically and computationally, is only a logarithmic factor compared to the method which has a-priori knowledge of the true $||w^*||$. So an improved method can, at best, remove this log factor. Optimizing these log factors is not the main focus of our work.

---

### Meta-Review · Area_Chair_f17M · 2022-08-30

**Recommendation:** Accept
**Confidence:** Certain

**Metareview:**

This work studies the problem of learning GLMs (generalized linear models) in the differentially private setting. The main results are nearly tight upper and lower bounds on the sample complexity of this task in a number of settings, including the smooth nonnegative case and the Lipschitz case. The reviewers appreciated the theoretical contribution and agreed that this paper is a good fit for the conference.

**Award:**

No

---

### Decision · Program_Chairs · 2022-09-14

Accept